



# Adaptation of the CIMEL-318T to Shipborne Use: Three Years of Automated AERONET-Compatible Aerosol Measurements Onboard the Research Vessel Marion Dufresne

Benjamin Torres[1], Luc Blarel[1], Philippe Goloub[1], Gaël Dubois[1], Maria Fernanda Sanchez-Barrero[1],
Ioana Elisabeta Popovici[1,2], Fabrice Maupin[2], Elena Lind[3], Alexander Smirnov[3,4], Ilya Slutsker[3,4],
Julien Chimot[5], Ramiro González[6,7], Michaël Sicard[8,9], Jean Marc Metzger[10], and Pierre Tulet[11]

[1]Univ. Lille, CNRS, UMR 8518 - LOA - Laboratoire d'Optique Atmosphérique, F-59000 Lille, France.
[2]R&D Department, Cimel Electronique, 75011 Paris, France.
[3]NASA Goddard Space Flight Center (GSFC), Greenbelt, MD20771, USA.
[4]Science Systems and Applications, Inc. (SSAI), Lanham, MD20706, USA.
[5]EUMETSAT, 64295 Darmstadt, Germany.
[6]Group of Atmospheric Optics, University of Valladolid (GOA-UVa), 47011, Valladolid, Spain.
[7]Laboratory for Disruptive Interdisciplinary Science (LaDIS), Universidad de Valladolid, 47011 Valladolid, Spain.
[8]Laboratoire de l'Atmosphère et des Cyclones (LACy), UMR 8105 CNRS, Université de La Réunion, Météo-France, 97744,
Saint-Denis de La Réunion, France.
[9]CommSensLab-UPC, Universitat Politècnica de Catalunya, Barcelona, 08034, Spain.
[10]OSU-R, CNRS/Université de La Réunion/Météo-France/IRD, UAR 3365, Saint-Denis, France.
[11]LAERO, UMR 5560 (CNRS, UT3, IRD), 31400, Toulouse, France.

**Correspondence:** Benjamin Torres (benjamin.torres@univ-lille.fr)

**Abstract.**

The Earth's oceans play a critical role in regulating the global climate and atmospheric processes, with marine aerosols significantly influencing weather patterns, air quality, and climate dynamics. Despite extensive land-based aerosol monitoring through networks like AERONET (AErosol RObotic NETwork), marine aerosol characterization remains a critical gap, due in part to the logistical challenges of conducting measurements in remote oceanic environments. To address this, robust, automated, and precise monitoring systems adapted for research vessels are essential.

This study reports on the first three years (July 2021 to June 2024) of continuous aerosol optical depth (AOD) measurements collected aboard the R.V. Marion Dufresne using a ship-adapted CIMEL 318-T automatic photometer in the frame of MAP-IO (Marion Dufresne Atmospheric Program - Indian Ocean) program. The dataset comprises over 25,000 quality-assured AOD measurements, primarily from the South-West Indian Ocean region, revealing mid-range AOD and Angström exponent values consistent with previous studies. The reliability and precision of the system were validated through dual-instrument comparisons conducted during the Amaryllis-Amagas/Transama campaign, yielding strong correlations (R > 0.96 for different wavelengths) and low root-mean-squared errors (RMSE < 0.01), within the expected error margins for AERONET ground-based sites, and benefiting from the continue tracking system implemented for ship-adapted version. Additionally, recurrent comparisons with the ground-based AERONET site at Saint-Denis (La Réunion) further confirm the system's accuracy, presenting good correlations despite differences in altitude and the greater influence of local urban aerosols in Saint-Denis.





Retrievals from spectral AOD and sky radiance data collected over the Indian Ocean during a biomass burning event (October 2023) demonstrate the feasibility of deriving detailed aerosol properties, including size distribution and optical characteristics, from shipborne platforms adapted for marine conditions, following the protocols of the AERONET standard algorithm. Observed SSA values, ranging from 0.88 to 0.95 with higher absorption at longer wavelengths, align with those recorded at the Saint-Denis site during the event and are consistent with expectations for a mixture of biomass burning (at the end of the dry season) and sea salt aerosols. These preliminary results underscore the potential of shipborne systems to provide comprehensive aerosol characterization in remote marine environments.

## 1 Introduction

Atmospheric aerosol optical studies, including radiative forcing analysis, aerosol-cloud interactions, remote sensing of the atmosphere, and global aerosol modeling, rely on precise information about AOD and aerosol absorption. Oceans, which cover approximately 70% of the Earth's surface (with 64% consisting of international waters), play a crucial role in the atmospheric system through sea-air interactions on both local and global scales. Specifically, oceans serve as a major source of sea spray aerosols and primary organic matter, such as chlorophyll-a (Gantt and Meskhidze, 2013). Sea spray aerosols, consisting of seawater droplets and dry sea salt particles (ionic species like sodium, potassium, etc.), are produced predominantly by bubble bursting from breaking waves (Blanchard and Woodcock, 1980). These particles range in size from 0.05 $\mu$m to 1 mm (Hoppel et al., 1990; O'Dowd et al., 1997; Porter and Clarke, 1997). The production of sea spray is influenced by factors such as wind speed, sea state, atmospheric stability, and seawater composition (O'Dowd and Smith, 1993; de Leeuw et al., 2011). Globally, sea spray aerosols are the dominant contributors to columnar AOD over oceans (Mahowald et al., 2006), and are the most widely dispersed natural aerosols with an estimated total flux of 4100 Tg yr$^{-1}$ according to the IPCC-2013 (Stocker et al., 2013). The contained primary marine organic matter is normally found in smaller particles under 200 nm (Leck and Bigg, 2008; Russell et al., 2010), with emission rates dependent on biological activity in ocean waters, estimated between 2 and 20 Tg yr$^{-1}$ (Facchini et al., 2008; Gantt et al., 2011). From a climatic perspective, the main interest in sea spray aerosols lies in their direct influence on the radiation budget due to their predominantly scattering nature (Gordon and Clark, 1981; Gathman, 1983; Haywood et al., 1999; Satheesh and Krishna Moorthy, 2005), as well as their indirect impact on cloud formation, dynamics, and life cycle, driven by their hygroscopic properties (e.g., Gunn and Phillips, 1957; Albrecht, 1989; O'Dowd et al., 1999; Van den Heever et al., 2006; Sandu et al., 2008).

Characterizing marine environments is particularly challenging due to the complexity of aerosol production and the variety of sources involved (Lewis and Schwartz, 2004). Marine atmospheric aerosols encompass both the natural components mentioned above emitted by the sea and aerosols transported from terrestrial sources, whether natural (e.g., desert dust) or anthropogenic (e.g., sulfates, nitrates, or biomass burning aerosols, see Prospero et al., 2002; O'Dowd and de Leeuw, 2007; O'Dowd et al., 2007). These aerosols contribute to the complex and dynamic nature of marine environments. For example, long-range transport of continental aerosols modifies the aerosol composition over the ocean, blending natural sea salt with desert dust or industrial emissions (Fitzgerald, 1991; O'Dowd et al., 2007). The generally low concentrations of marine aerosols make their climatic



effects highly sensitive to small changes. As Koren et al. (2014) suggested, even minor variations can significantly influence cloud formation and radiative forcing, often resulting in negative forcing (cooling) by promoting convective cloud development.

Pristine marine regions, such as the Southern Indian Ocean and the South Pacific, are particularly valuable for understanding these processes because there is minimal human influence on them (Hamilton et al., 2014). Studying aerosols in these regions also provides information on preindustrial meteorological conditions, which serve as a baseline to evaluate the contribution 55 of natural emissions to climate change. Furthermore, recent findings highlight the role of marine aerosols in climate feedback mechanisms, with processes such as marine cloud brightening (Alterskjær and Kristjánsson, 2013), illustrating the sensitivity of radiative forcing to aerosol-cloud interactions (Twomey, 1977). However, the efficacy of such processes is influenced by meteorological conditions, background aerosol levels, and particle properties, resulting in a wide range of potential impacts on radiative forcing, estimated between $-5.4$ and $-0.8$ W m$^{-2}$ (Intergovernmental Panel on Climate Change (IPCC), 2021, 60 mostly based on the analysis by Pringle et al., 2012).

Despite advances in understanding marine aerosols, substantial uncertainties continue to exist due to their strong dependence on sources, emissions, and interactions with clouds. Variability in sea spray concentrations, driven by changes in wind speed, sea state, and sea ice cover, adds further complexity (Struthers et al., 2011). Climate projections suggest that these emissions could increase or decrease under changing conditions, with potential implications for aerosol-cloud-climate feedbacks (Jones 65 et al., 2007). Addressing these uncertainties requires comprehensive observational datasets to refine models, improve estimates of aerosol sources, and better quantify their climatic impacts. Systematic measurements of aerosol optical properties in maritime environments, coupled with robust statistical analyzes, are therefore essential to advance our understanding of regional marine aerosol climatologies, their trends, and their global effects.

Marine aerosol optical properties can be measured using passive remote sensing instruments from spaceborne, airborne, or 70 shipborne platforms. Spaceborne observations offer a global, long-term perspective on marine aerosol conditions, with aerosol properties retrieved by analyzing spectral radiances collected by their sensors from solar reflection off the Earth's surface and transmission through the atmosphere. In this context, satellite data retrievals either rely on assumptions about surface reflectance or employ a combined retrieval of aerosol and surface properties (Dubovik et al., 2011; Sayer et al., 2018), both of which introduce non-negligible errors. The accuracy and level of detail of aerosol products derived from satellite data vary 75 depending on sensor performance and capabilities (e.g., multi-wavelength, multi-angular, and polarization measurements). For the most advanced sensors, AOD uncertainties at 550 nm typically start at 0.04–0.05 over oceans and around 0.1 over land (see, for example, validation studies in Gupta et al., 2018; Dubovik et al., 2019; Chen et al., 2020).

In contrast, AOD from Earth-based measurements is obtained through direct irradiance measurements of the Sun (or Moon) using precise photometers and applying the well-known Beer–Bouguer–Lambert law (Shaw, 1983), which describes the at-80 tenuation of solar or lunar light as it passes through the atmosphere. The associated uncertainty is around 0.01 to 0.02 and is mainly due to errors in the calibration coefficient in the different channels (Eck et al., 1999; Holben et al., 2006; Giles et al., 2019), a level significantly lower than that of satellite-derived AOD products. The high precision of Earth-based measurements is particularly valuable for improving our understanding of regional marine aerosol properties and their climatologies, while also refining uncertainty estimates for satellite aerosol products over oceans. These satellite products are rarely validated



with Earth-based data due to a critical lack of ground observations in oceanic regions. In most validation studies, available data usually come from island stations or coastal sites, rather than from pure open-ocean environments. Additionally, this lack of Earth-based data in oceanic areas also complicates the validation of aerosol transport models, which typically rely on ground-based data to extend coverage of detailed aerosol properties beyond fixed ground-based sites. This gap in validated marine aerosol data hinders a comprehensive understanding of general aerosol dynamics and impedes accurate global climate

predictions.

One reason for the lack of data over oceans is that most aerosol ground-based networks have historically concentrated on studying aerosol properties over land. For example, the well-known AERONET network (Holben et al., 1998, 2001) comprises more than six hundred land-based sites, with limited information over the oceans, primarily covering islands. However, not all areas of the World Ocean can be studied from islands; aside from space and airborne sensors, ships are the only platform where

measurements can be obtained. Consequently, despite challenges such as platform mobility and the harsh marine environment, including exposure to salt and high humidity, shipborne sun photometer observations have advanced significantly over the past 20 years. The largest long-term aerosol observation network in the ocean is the Maritime Aerosol Network (MAN) (Smirnov et al., 2009), a component of AERONET. MAN inherits its legacy from the NASA SIMBIOS (Sensor Intercalibration and Merger for Biological and Interdisciplinary Oceanic Studies) program (Fargion et al., 2000, 2003; Knobelspiesse et al., 2004).

MAN provides a unique data set of AOD, the Angström exponent (as defined by Ångström (1929) and commonly calculated within AERONET using a least-squares method over the 440–870 nm wavelength range), and precipitable water vapor (PWV) in the ocean, spanning from the Arctic to Antarctica.

MAN exploits the advanced AERONET calibration facilities and processing schemes and relies on many logistical and scientific developments from the AERONET Project. The MAN web-based public data archive is available from the AERONET

website. MAN represents an important strategic sampling initiative, and ship-borne data acquisition complements island-based AERONET measurements. MAN started collecting data over the oceans in November 2006 and has since made significant progress in data collection and archival. With more than 750 cruises completed and ongoing (and many more planned), the MAN database continues to grow, enhancing our knowledge of spectral AOD variation over the oceans. The ultimate objective is to advance the fundamental scientific understanding of aerosol optical properties globally through highly accurate and

standardized measurements, providing a basis for evaluation and inter-comparison of aerosol optical depth (AOD) retrievals from various spaceborne sensors and outputs of the global aerosol transport models. However, unlike AERONET ground-based standard instruments, the standard device of MAN is the hand-held Microtops II Sun photometer, which requires manual operation, making it less suitable for continuous and unattended measurements. Furthermore, it cannot provide aerosol optical and microphysical properties obtained through the AERONET aerosol retrieval algorithm (Dubovik and King, 2000; Dubovik

et al., 2000, 2002b, 2006; Sinyuk et al., 2020) due to the lack of sky radiance measurements (Smirnov et al., 2009).

For these reasons, the development of a ship version of the automated standard instrument of the AERONET Network ground-based sites (CIMEL Sun photometer) has been a primary objective within the framework of the Agora Lab (CIMEL Electronique Company and Université de Lille in a "laboratoire commun de recherche", https://www.agora-lab.fr, last access: 31 December 2024) during recent years. The objective is to obtain automatic spectral AODs (340 to 1640 nm), spectral





downward atmospheric radiances (380 to 1640 nm), and column integrated water vapor information on ship-borne platforms, achieving identical measurements and protocols as the ground-based photometers. This strategy ensures consistency and accuracy across the network and should enable the provision of comprehensive aerosol microphysical and optical properties, including size distribution, scattering phase function, and single scattering albedo, from vessels operating at sea.

The key aspect of adapting the CIMEL photometer for ship-based use is the integration of the boat's position and movement-
related information, collectively referred to as attitude, which includes heading, pitch, and roll. This integration has only been possible since the modernization introduced in the latest version of CIMEL photometers: the CIMEL CE318-T (Barreto et al., 2016). This new version includes a GPS receiver that automatically determines the location, unlike previous CIMEL versions, where the position had to be manually entered during installation. For the ship version, the GPS data is currently provided every 5 minutes, compared to once per day in standard ground-based usage. Additionally, the photometer's orientation
during installation is more flexible thanks to an automatic azimuth correction system that operates during the initial setup. The electronic control box has also been upgraded and now compensates for the robot's movements by accounting for the boat's attitude information. Another significant improvement is the implementation of continuous Sun (or Moon) tracking during direct Sun (or Moon) measurements. In ground-based sites, Sun tracking is performed only once, prior to acquiring spectral direct Sun measurements (using 9 filters, in a process repeated three times to produce a triplet sequence, as explained in
subsection 2.2). In contrast, on ships, tracking is continuous to maintain the instrument's Sun pointing throughout the AOD measurement process, ensuring accurate AOD measurements (also obtained spectrally and as a triplet).

The first tests to perform automatic measurements with the CIMEL CE318-T at sea were conducted on the Research Vessel (R.V.) Kommandor-Iona during the AQABA campaign (Eger et al., 2019; Kaskaoutis et al., 2023; Pfannerstill et al., 2019) around the Arabian Peninsula during the summer of 2017. To obtain the attitude information of the vessel, previous experi-
ence from developing photometry measurements on moving platforms was leveraged. Specifically, the magnetic compass for tracking the plane's attitude, included in the PLASMA airborne photometer developments (Karol et al., 2013), was used in this first attempt as a magnetic declination calculator. The CIMEL software was adapted to integrate this attitude information though only to perform direct Sun measurements. The system worked and produced the first ship-borne AOD automatic CIMEL CE318-T measurements on the trajectory from the south of France to Kuwait (Unga et al., 2019). However, a recurrent
malfunction was experienced in the last part of the campaign since the magnetic compass system required frequent calibration to account for the magnetic environment. Note that this frequent calibration is feasible for short campaigns, as it is performed before almost every flight where PLASMA is involved, but impractical for long-term fully automated measurement systems. Moreover, magnetic declination information should be provided to ensure the proper functioning of the compass calculation system, which was not included in the initial software. This campaign also highlighted the need for the "airshield" (name given
to the anti sea spray system used, based on a continuous air supply system), as water ingress in the optics due to sea spray was observed. Furthermore, a more robust rain sensor was identified as necessary, as the CIMEL CE318-T standard one became oxidized due to sea conditions in combination with regular high temperatures in the Red Sea. Furthermore, the need for a wind speed monitoring device was emphasized to halt measurements when necessary, as strong winds at sea could cause excessive instrument movement or allow water ingress into the sensors due to high waves frequently encountered during storms at sea.





The second attempts took place during the OCEANET transatlantic campaigns (PS113, PS116, and MOSAIC/Arctic) with the R.V. Polarstern (Yin et al., 2019). In those campaigns, several modifications were introduced. First, a pump system was implemented to direct dry air from inside the boat to the CIMEL collimator to counteract the effects of sea spray. The system successfully avoided the input of sea spray into the CIMEL CE318-T optics, but on some occasions, condensation problems arose depending on the differences between external humidity and temperature and those inside the boat. Second, an optical

rain sensor replaced the standard resistive rain sensor of the CIMEL CE318-T, as it is not prone to corrosion and better suited for marine conditions. Third, an anemometer was installed to monitor wind speed and automatically disable the CIMEL CE318-T during strong winds. Fourth, the basic electronic card-based attitude measurement system (magnetic declination calculator) was replaced by a SIMRAD-H60 commercial navigation compass, which includes a dual GPS antenna and a declination-corrected magnetic compass. With all these changes, the quality andber of AOD products imprimproved greatlypared to the first

campaign, and Yin et al. (2019) presented a first comparison of these automated AOD measurements with a MAN Microtops-II instrument, shoshowing root mean squaredferences of 0.015, 0.013, 0.010, and 0.009 at 380, 440, 500, and 870 nm channels respectively, and correlations of 0.99 for all alls.

However, despite testing two versions of the SIMRAD-H60 system during the campaigns on the R.V. Polarstern (MO-SAIC/Arctic campaign), attitude errors ranging between $2$–$5°$ occasionally complicated the Sun acquisition, as the field of

view of the CIMEL 318-T tracking system is approximately $3°$. This solution was shown inefficient for our proposed automated independent final solution, where the instrument can be deployed for several months at sea without continuous supervision.

The last improvements leading to the current version were introduced in 2020 (all the final technical solutions comprising the current system will be described in detail in subsection 2.1). During the Sea2Cloud campaign aboard the R.V. Tangaroa, the SIMRAD-H60 system was replaced with the ABX-Two inertial GPS unit, ensuring a root mean square error (RMSE)

of $0.07$–$0.1°$ for heading and $0.13$–$0.2°$ for pitch and roll. [1] The system functioned correctly after some initial adjustments to the position of the antennas. Unfortunately, only a few AOD measurements were obtained since the campaign started on March 1, 2020, and was canceled by the end of the month due to the COVID-19 pandemic. In late 2020, additional tests were conducted on fishing boats (small vessels of around 20 meters in length) off the coast of North of France. The fishing boats conducted daily round trips, allowing for numerous tests and on-site modifications. During these tests, the final airshield system

was installed outside, effectively preventing condensation issues. Additionally, the CIMEL firmware was adapted to perform automatic radiance (Almucantar) measurements for the ship-version.

By early 2021, the goal was to install a fully AERONET-compatible photometer on the R.V. Marion Dufresne (a ship in the French oceanographic fleet operated by Ifremer) in January as part of the MAP-IO program (Tulet et al. (2024), more information on the boat and MAP-IO project in subsection 2.3). This instrument included all the necessary adaptations for

sea operations and the final version of the software, capable of performing both direct and radiance measurements. However,

---

[1] The specifications of the ABX-Two inertial GPS unit indicate an RMSE of $0.1°$ per meter of baseline (distance between the antennas) for heading and $0.2°$ for pitch and roll. Since the final setup of antenna separation on board the R.V. Marion Dufresne ranges from 1.5 to 2 meters, the associated errors fall within these stated values. Internal tests conducted on fixed platforms confirm these specifications. In contrast, the specifications for the SIMRAD-H60 system indicate a root-mean-square (RMS) error of $2°$ (with $68\%$ of values below this threshold); however, similar internal tests have revealed larger errors in practice, reaching up to $5°$ as stated in the text.





during the installation, a calibration issue with the instrument was discovered. Consequently, another photometer with an older version of the software was installed and provided the first measurements onboard the R.V. Marion Dufresne. This instrument, which operated from January to March 2021, routinely performed automatic AOD measurements, and some manual radiance measurements were taken (as it lacked the updated system for automatic sky-radiance measurements). The initial instrument
was recalibrated, and on 1 July 2021 the first fully compatible instrument with AERONET on a seaboat was successfully installed onboard the R.V. Marion Dufresne, marking a significant milestone in achieving 100% AERONET compatibility at sea. The analysis of these data is the primary objective of this paper and is presented in detail in sections 3 and 4.

## 2  Instrument, site and data treatment

### 2.1  Instrument: Sea-adapted CIMEL 318-T photometer

To effectively monitor aerosol properties over the ocean, a specialized adaptation of the standard ground-based AERONET instruments was required. This section details the development and components of the sea-adapted CIMEL 318-T photometer and the associated data processing methods. A scheme of the photometer adapted for boat conditions is presented in Figure 1.

The prototype was designed as a modular solution, with the CIMEL 318-T photometer as the core component (represented in the middle part of Figure 1). This instrument is the most recent standard instrument in the AERONET network and is
originally designed for ground-based installations. The second key component is the inertial GPS unit Trimble ABX-Two (on the right part of Figure 1), which provides real-time attitude information, including heading, roll, and pitch with an estimated uncertainty $< 0.2°$. The unit continuously transmits the ship's attitude data to one of the photometer's communication ports. Combined with geo-localization from the photometer's internal GPS, this allows the system to continuously determine the precise position of the Sun and the Moon, ensuring accurate Sun or Moon pointing. Once the Sun or Moon enters the
tracking system's field of view, the photometer switches into tracking mode, just like a regular AERONET instrument, and subsequently performs direct-Sun measurements. However, unlike conventional ground-based photometers, where tracking is performed once immediately before each direct-Sun measurement, the shipborne photometer maintains continuous tracking in parallel with the direct measurement. This ensures that even as the vessel moves, the instrument remains locked onto the Sun, maintaining precise alignment for accurate data acquisition. Additionally, the ABX-Two generates continuous positioning
data, that together with attitude information are recorded every second on a dedicated PC. This PC also collects data from the photometer and also ensures connectivity with the ship's communication system (Wi-Fi/cable), facilitating remote data access.

The third component of the system is the air pumping unit (left side of Figure 1), which prevents contamination of the optics by sea spray. A continuous airflow is injected at the base of the collimator to create an overpressure, preventing both the deposition of sea spray on the optics and the intrusion of particles into the collimator. The airflow is delivered through a
butyl and flexible white hose connected to a pump, ensuring a consistent and sufficient air supply. The flexible hose minimizes pressure loss while allowing free movement of the robotic tracking system. A filter installed at the air intake (Box1) ensures that only clean air flows through the system. To maintain optimal optical conditions, the pump operates continuously (even if



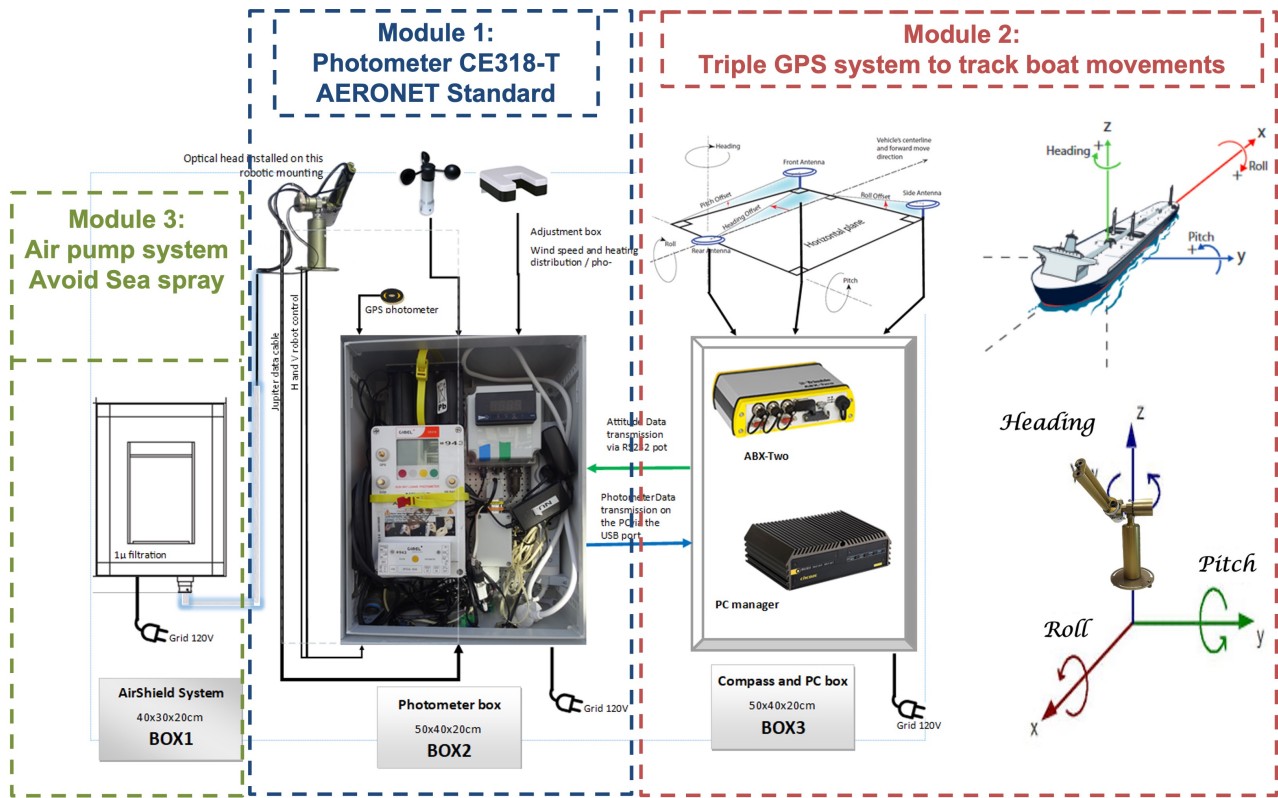

**Figure 1.** Scheme of the photometer adapted for boat conditions. The central part of the figure shows the triple version of the CIMEL CE318-T photometer, the latest standard instrument in the AERONET network, originally designed for ground-based installations. The right side displays the Trimble ABX-Two inertial GPS unit, which provides attitude data (heading, roll, pitch) and continuously transmits this information to the photometer's communication ports. The left side of the figure illustrates the air pumping unit, which supplies clean, compressed air to the base of the collimator, preventing sea spray deposition. A weather system, including a non-corrosive optical rain sensor and an anemometer, is also integrated to halt operation during rain or high winds. Source: Pictures of Module 3 taken from the User Guide of the Trimble ABX-Two.

the photometer is stopped) to prevent salt deposition on the optics or within the collimator. The pump can, however, be stopped when the photometric head is removed.

Finally, the system features a weather unit that replaces the standard CE318-T resistive wet sensor, which was prone to corrosion in marine conditions. This unit includes an optical rain sensor and an anemometer to monitor wind speed and stop measurements during extreme conditions, protecting the instrument from strong winds and high waves. Unlike standard AERONET ground-based instruments, which pause only for rain, this system halts measurements for both rain and high winds. The weather unit connects to the photometer through the standard humidity sensor port.





## 2.2 Data Treatment and availability

The advantage of using the standard AERONET photometer CE318-T version for the shipborne system lies in its full compatibility with AERONET calibration, which allows for the extension of quality control/quality assurance (QC/QA) procedures to all measurements. This compatibility significantly simplifies the data processing workflow. The term "AERONET compatible" implies that the shipborne photometers follow the same measurement protocols and schedules as the ground-based instruments, use identical filters, and undergo the same data processing procedures. Additionally, the calibration of SUN, MOON, (Sun and Moon direct measurements) and SKY measurements (sky radiances) is performed in the same manner as for terrestrial photometers, ensuring consistency and accuracy across the network.

Once the raw data from R.V. Marion Dufresne are collected, they are transmitted via satellite to the server of the PHOTONS CNRS National Observation Service (University of Lille) and then forwarded to the NASA server. The data treatment and the assignment of Level 1.0 or Level 1.5 follow the same protocols as those described in the AERONET Version 3 processing system (Giles et al., 2019), identical to those applied at regular fixed ground-based sites.

The achievement of Level 1.0 is mainly based on the analysis of the instrument's electronic signal, recorded as digital number (DN) values corresponding to direct Sun (or Moon, for nighttime AOD) irradiance measurements, or the sensor head temperature (Giles et al., 2019). Digital count anomalies can prevent measurements from achieving Level 1.0 and may arise from electronic issues such as extreme battery voltages, malfunctioning amplifiers, or loose connections within the internal control box, though most of these issues have been mitigated with the new CIMEL CE318-T photometer. Thus, in newer versions of the photometer, the primary factor limiting the achivement of Level 1.0 is incorrect Sun (or Moon) pointing, either due to cloud obstruction or tracking system errors. The AOD is derived from DN measurements using the extraterrestrial calibration coefficient (i.e., the DN corresponding to extraterrestrial irradiance) and applying the Beer–Bouguer–Lambert law (Shaw, 1983; Holben et al., 1998). In AERONET, AOD has historically been determined from three sequential measurements of the Sun's (or Moon's) irradiance taken within a one-minute interval (Smirnov et al., 2000). To ensure measurement validity, several quality control criteria are applied, including a minimum signal threshold requiring at least 100 counts in the infrared channels (870 and 1020 nm). Additionally, any raw signal lower than the extraterrestrial signal (calibration factor) divided by 1500 (corresponding to a total optical depth of 2 multiplied by an air mass of at least 4 or SZA= 75°) results in the rejection of the corresponding channel. Furthermore, the DN triplet variance criterion (described in section 3.1.3 of Giles et al. (2019)) is applied, whereby if the relative RMSE of the DN triplet exceeds 16%, the entire observation is discarded.

Approximately 30% of the total SUN/MOON triplets from the shipborne photometer aboard the R.V. Marion Dufresne achieved Level 1.0 during the analyzed period (July 1, 2021 – June 30, 2024). Since the R.V. Marion Dufresne regularly operates near La Réunion, the AERONET site "REUNION_ST_DENIS" (hereafter Saint-Denis), located on the roof of the University of La Réunion at 93 meters above sea level, provides a relevant ground-based reference. Given that the transition to Level 1.0 is strongly influenced by instrument location and local climate conditions (obstruction due to thick clouds) this comparison offers valuable context. For the same period, approximately 47% of the measurements at Saint-Denis reached Level 1.0. This difference is reasonable, considering the additional challenges faced by shipborne measurements, primarily the vessel's





motion and rotational movements. Additionally, part of the data loss can be attributed to installation-specific constraints. Un-
like AERONET ground-based stations, which are typically installed in locations with a fully unobstructed 360-degree horizon
to maximize measurement opportunities, shipborne platforms are subject to obstructions from onboard structures, masts, and
operational equipment. As shown in Figure 1 of Tulet et al. (2024), the positioning of the photometer aboard the R.V. Marion
Dufresne places it near a platform structure, further reducing the fraction of successful Level 1.0 measurements. Taking these
constraints into account, the 30% success rate compared to 47% at Saint-Denis highlights the robustness of the shipborne
system and its capability to maintain tracking in a high percentage of cases, even under challenging conditions. This result
underscores the effective operation of the tracking mechanism and its ability to ensure stable performance despite the dynamic
nature of maritime environments.

Although the DN triplet variance criterion is sensitive to clouds with large spatial-temporal variations in optical depth,
including optically thick clouds such as cumulus, achieving Level 1.5 (cloud-screened AOD) requires a more refined screening
process to detect thinner and more homogeneous clouds, such as cirrus. In AERONET Version 2, the primary cloud screening
method relied on the AOD triplet variability test, which assessed AOD fluctuations within each measurement triplet. A triplet
was considered valid if the difference between the maximum and minimum AOD remained below 0.02 across all wavelengths.
In cases of high aerosol loading (such as biomass burning or extreme haze events), a more flexible threshold was applied,
allowing a maximum variability of 0.03· AOD, with the less restrictive criterion taking precedence when necessary. Version 3
(Giles et al., 2019) introduced further refinements to the triplet variability test. On one hand, the spectral criterion was adjusted,
limiting the analysis to only three wavelengths (675, 870, and 1020 nm) instead of all available channels. This change prevents
unnecessary exclusions of fine-mode aerosol data, where natural variability at shorter wavelengths can be higher (Eck et al.,
2018). At the same time, the variability threshold was made more stringent, requiring that AOD triplet variability remains
below the maximum of 0.01 or 0.015· AOD across these three wavelengths simultaneously. This stricter threshold enhances
cloud screening effectiveness while ensuring that stable fine-mode aerosol conditions are retained in the dataset.

Furthermore, the temporal AOD variation thresholds were refined for improved screening accuracy (details in Table 2 of
Giles et al., 2019), and an angular radiance test was implemented to detect cirrus clouds based on the analysis of forward
scattering in the solar aureole[2]. Cirrus clouds, composed of large ice crystals, produce a distinctive forward scattering peak,
which can be identified by evaluating the shape of the solar aureole at small scattering angles ($3.2°$–$6.0°$). The analysis relies
on assessing the curvature and slope of the aureole radiances using a logarithmic regression. Specifically, a low curvature value
at the smallest scattering angle and a high slope across the measured angles are indicative of enhanced forward scattering,
suggesting the presence of cirrus clouds. These advancements, along with others highlighted by Giles et al. (2019), further
refine the cloud screening process in Version 3, enabling a more effective removal of cloud-contaminated data and enhancing
the quality of Level 1.5 AOD measurements. Consequently, the Level 1.5 AOD data collected aboard the R.V. Marion Dufresne

---

[2]For T-Model photometers, cirrus cloud detection can be performed using a new dedicated measurement scenario called the cross-curvature scan (CCS).
This scan, conducted at 1020 nm, measures aureole radiance at specific scattering angles before each solar triplet measurement. In older CIMEL versions,
cirrus detection relies on analyzing almucantar radiance measurements, with all spectral AOD data removed within 30 minutes of the sky measurement if the
curvature and slope do not accomplish the thresholds defined in Giles et al. (2019). Although the shipborne photometer is a T-Model instrument, CCS scans
are disabled in the current firmware version installed, and cirrus detection is currently performed using standard sky radiance analysis.



and presented here have undergone rigorous cloud screening, ensuring that the retained Sun and Moon measurements comply with AERONET's quality standards. The percentage of Level 1.0 data successfully advancing to Level 1.5 is approximately 70%, while at the Saint-Denis site, this percentage is slightly higher, reaching nearly 80%. This 10% difference could be attributed to occasional tracking imperfections during triplets that meet the 16% DN variability criterion but fail to satisfy the 2% AOD variability threshold in AERONET's cloud-screening procedure.

However, while Level 2 quality-assured AOD data—recommended for publication and use in various atmospheric applications (Giles et al., 2019)— can be routinely achieved for ground-based observations, shipborne data are currently limited to Level 1.5, as additional validation steps are still required to account for the unique challenges of moving platforms. Further validation studies, such as those presented by Yin et al. (2019) and the present study, are essential to establish the reliability of CIMEL 318-T shipborne data within the AERONET network.

Unlike ground-based photometers, not all raw data received from the shipborne photometer are automatically processed. Following an approach similar to the MAN procedures, only data from direct Sun or Moon measurements—such as spectral AOD, water vapor content, and the Ångström Exponent—are processed when the vessel is officially on a mission. This prevents unnecessary processing of extended periods when the vessel is docked at port. However, some data collected while the boat is docked are still processed, particularly during testing missions where the vessel departs and returns to port daily. Additionally, on certain days within declared testing periods, the vessel may remain at the dock while data continue to be processed. In the three-year analysis presented in this paper, approximately 20% of the processed data correspond to periods when the vessel was within the port area[3]. This dataset includes both daytime and nighttime measurements and has been incorporated into the analysis presented in subsection 3.1.

The determination of downward-atmospheric calibrated radiances from sky measurements, following the almucantar geometry, is processed during declared mission periods. Although the T-Model version of the CIMEL photometer is capable of performing hybrid sky scans —combining simultaneous azimuthal and zenith angle movements to provide intermediate data between almucantar and principal plane geometries (Sinyuk et al., 2020)— the photometer installed aboard the R.V. Marion Dufresne is currently limited to almucantar scans due to the specific software configuration. This limitation, while restricting hybrid scan capabilities, still enables the instrument to meet AERONET compatibility standards and produce high-quality sky radiance measurements. As with other regular AERONET instruments in the network, almucantar scans are executed at fixed elevation angles equal to the solar elevation, with $\pm 180°$ azimuthal sweeps made sequentially at four wavelengths (440, 675, 870, and 1020 nm). These scans are typically performed during the morning and afternoon at specific optical air masses of 3.8, 2.9, 2, 1.7, and 1.4, corresponding to solar zenith angles ($\theta_S$) of 75, 70, 60, 54, and 45°, respectively.

Almucantar sky radiance data, combined with spectral AOD measurements at identical wavelengths, can be used in the AERONET inversion algorithm developed by Dubovik and King (2000) and later expanded by Dubovik et al. (2006). This algorithm retrieves optically equivalent, column-integrated volume size distributions and aerosol refractive indices, which are further utilized to derive secondary aerosol properties such as single scattering albedo (SSA), the asymmetry parameter, and phase functions. For the first time, this methodology has been applied to perform such retrievals from a shipborne platform

---

[3]The ship does not always remain in the same position within the port; the port area is defined as approximately one square kilometer.





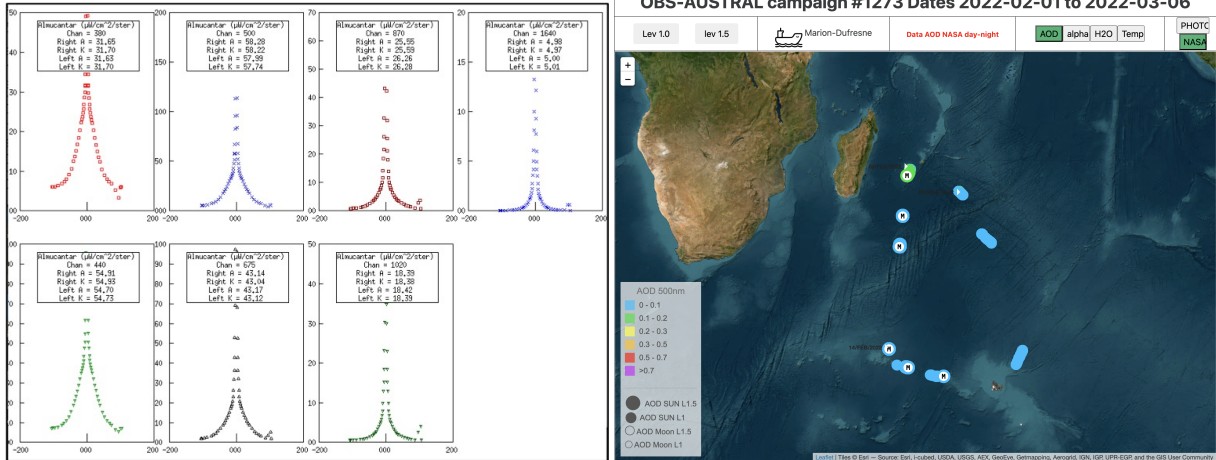

**Figure 2.** Left: Example of calibrated radiance measurements following an almucantar scenario on October 17, 2023, onboard the R.V. Marion Dufresne from Demonstrat (AERONET internal system). Right: Visualization of $AOD_{440}$ during the OBS-AUSTRAL campaign from January to March 2022, as displayed on the PHOTONS system platform (https://mobile.photons.univ-lille.fr/, last access: 31 December 2024).

within the AERONET network. However, retrievals for shipborne platforms are not yet performed automatically. Processing

has been done for selected periods of interest, and the first results are discussed in this study (section 4). Although all ground-based quality control criteria, such as the symmetry of the almucantar scans, are considered in these initial inversions, additional quality criteria for shipborne retrievals are being analyzed. This includes setting extra thresholds for the angles of the ship's movements during sky radiance scenarios (or accounting for their effects), improving the model of sea water reflectances, and other factors.

The aforementioned AOD, water vapor, calibrated radiance, and initial aerosol microphysical and optical properties from the first AERONET aerosol retrievals are currently available internally through Demonstrat, an internal AERONET software used for aiding calibration, technical control of equipment, and data review. An example of calibrated radiance measurements following an almucantar scenario on October 17, 2023, onboard the R.V. Marion Dufresne is presented on the left in Figure 2. These data will soon be accessible on the general AERONET website (https://aeronet.gsfc.nasa.gov, last access: 31 December

2024). This delay is due to the technical modifications required for the visualization software to handle data from moving platforms, as opposed to ground-based sites. In addition to AERONET processing and archiving, a near-real-time visualization system has been developed within the PHOTONS system (https://mobile.photons.univ-lille.fr/, last access: 31 December 2024). By selecting the mission and the boat, it is possible to visualize and download the corresponding AOD data (see the right panel of Figure 2, which shows the AOD at 440 nm ($AOD_{440}$) during the OBS-AUSTRAL campaign from January to March 2022).





## 2.3 R.V. Marion Dufresne and MAP-IO program

The R.V. Marion Dufresne is a state-of-the-art oceanographic research vessel operated by Terres Australes et Antarctiques Françaises (TAAF) for 120 days per year and by the French Research Institute for Exploitation of the Sea (Ifremer) for the remaining 220 days. Designed for long-range, multidisciplinary scientific campaigns, it is equipped with advanced laboratories and cutting-edge technologies, making it an ideal platform for atmospheric aerosol studies in marine regions.

The MAP-IO program, launched in early 2021, aims to address the critical lack of atmospheric and oceanographic observations in the Southern Indian and Southern Oceans—key regions in global climate regulation that remain among the least studied. By the end of 2024, after approximately 1100 days at sea, MAP-IO will have deployed 17 scientific instruments aboard the R.V. Marion Dufresne, collecting unprecedented data on atmospheric aerosols, greenhouse gases, ultraviolet radiation, and water vapor, along with high-resolution phytoplankton observations in surface waters. The primary objective of MAP-IO is to assess the feasibility of establishing a permanent marine observatory aboard the Marion Dufresne, integrating it into international atmospheric and oceanographic monitoring networks. The collected data provide valuable insights into aerosol distribution and optical thickness, as well as seasonal variability in marine aerosols, greenhouse gases, and ocean-atmosphere interactions in these climatically sensitive regions. MAP-IO represents a strategic initiative to bridge observational gaps, enabling more accurate assessments of regional aerosol dynamics and their climatic impacts while supporting the development and validation of aerosol retrieval algorithms for marine environments, such as those explored in this study. In 2025, MAP-IO was accredited as a French CNRS National Instrument, securing its operation until at least 2030. All observations collected under the MAP-IO program are available through the AERIS atmospheric data center (https://www.aeris-data.fr/catalogue-map-io/, last accessed: 31 December 2024) and the SEANOE ocean data center (https://doi.org/10.17882/89505, last accessed: 31 December 2024). More information can be found at www.mapio.re (last accessed: 31 December 2024) or in Tulet et al. (2024).

## 3 Data Analysis

### 3.1 Aerosol Optical Depth

This section presents the AOD measurements obtained aboard the R.V. Marion Dufresne over a three-year period, from July 1, 2021, to June 30, 2024. Table 1 summarizes the key statistical metrics for this dataset, including the total number of data points per channel, the mean and standard deviation, and the percentiles at $25\%$, $50\%$, $75\%$, and $95\%$, along with the minimum and maximum values.

The total number of measurements containing at least one wavelength at Level 1.5 is $25,602$, with $17,293$ recorded during the day and $7,735$ at night using Moon light. Only the 500 and 870 nm channels have all their measurements validated at Level 1.5. It is important to note that certain Level 1.0 criteria (outlined in subsection 2.2 and detailed in Giles et al., 2019) apply to individual channels and can result in the exclusion of specific wavelengths. Since Level 1.5 processing requires a valid Level 1.0 measurement, any data failing Level 1.0 criteria cannot advance to Level 1.5. Furthermore, the transition from Level 1.0 to Level 1.5 may lead to the removal of either the entire spectral AOD measurements (mainly due to cloud screening



**Table 1.** Summary statistics of the AOD data collected aboard the R.V. Marion Dufresne from July 1, 2021, to June 30, 2024. The table includes the total number of data points for each channel, the mean value and its standard deviation, and percentiles at 25%, 50%, 75%, and 95%, along with the minimum and maximum values.

| Value | Count | Mean | Std | Min | 25% | 50% | 75% | 95% | Max |
|---|---|---|---|---|---|---|---|---|---|
| $AOD_{340}$ | 17302 | 0.116 | 0.118 | 0.007 | 0.059 | 0.086 | 0.120 | 0.308 | 1.009 |
| $AOD_{380}$ | 17314 | 0.112 | 0.104 | 0.012 | 0.062 | 0.085 | 0.117 | 0.284 | 0.888 |
| $AOD_{440}$ | 25059 | 0.093 | 0.075 | -0.005 | 0.054 | 0.075 | 0.103 | 0.207 | 0.717 |
| $AOD_{500}$ | 25062 | 0.084 | 0.063 | 0.009 | 0.051 | 0.070 | 0.095 | 0.182 | 0.584 |
| $AOD_{675}$ | 25061 | 0.063 | 0.041 | 0.004 | 0.039 | 0.054 | 0.075 | 0.136 | 0.384 |
| $AOD_{870}$ | 25062 | 0.054 | 0.031 | 0.003 | 0.034 | 0.048 | 0.067 | 0.115 | 0.328 |
| $AOD_{1020}$ | 24995 | 0.049 | 0.029 | 0.001 | 0.029 | 0.043 | 0.061 | 0.105 | 0.341 |
| $AOD_{1640}$ | 24765 | 0.036 | 0.026 | -0.006 | 0.019 | 0.030 | 0.047 | 0.082 | 0.367 |
| Ang. Exp. | 25062 | 0.754 | 0.400 | -0.823 | 0.459 | 0.691 | 1.024 | 1.484 | 2.546 |

criteria such as AOD variability or solar aureole curvature analysis, as described in Subsection 2.2) or individual wavelengths[4] from the AERONET AOD database. The few exclusions in the 440 and 675 nm channels (only 3 and 1 points, respectively) are attributed to these criteria.

A significant portion of the missing data in the remaining channels can be attributed to nighttime observations, though some exclusions also result from the aforementioned Level 1.0 and Level 1.5 criteria. For the 340 nm and 380 nm channels, AOD estimation from Moon measurements systematically excludes these wavelengths due to the low incoming lunar irradiance in this spectral range (Barreto et al., 2013, 2016). As a result, none of the 7,735 nighttime Level 1.5 measurements include these two channels. For the near-infrared channels (1020 nm and especially 1640 nm), the lunar irradiance signal is lower than in the visible range. This limitation is particularly pronounced during the first and last quarters of the lunar cycle (the first and last days with AOD night observations), when the received light is minimal. On one hand, this increases the measurement uncertainty for these channels compared to the visible range on these moon quarter days (Barreto et al., 2016). On the other hand, if the signal is too weak, it may fail the Level 1.0 filters related to minimum signal requirements, either due to the minimum digital number threshold or because the signal is lower than the extraterrestrial signal divided by 1500, both of which apply to individual wavelengths. As a consequence, some moon quarter days lack AOD data at 1020 nm and 1640 nm, even when measurements are available for visible channels. This explains most of the nearly 300 missing data points at 1640 nm and the 67 missing at 1020 nm.

Table 2 presents the same statistics as Table 1, but computed using daily mean AOD values. This analysis includes a total of 344 days of observations, of which 60 contain only nighttime measurements, meaning the 340 and 380 nm channels have valid data for 284 days. This results in an overall observation frequency of approximately one out of every three days over the

---

[4]For instance, criteria such as "Aerosol optical depth spectral dependence" and "Large aerosol optical depth triplet variability" (subsections 3.3.5 and 3.3.6 in Giles et al. (2019)) applied specifically to individual wavelengths.





**Table 2.** Statistical summary of daily mean AOD values at different wavelengths obtained aboard the R.V. Marion Dufresne from July 1, 2021, to June 30, 2024. The table presents the number of valid daily averages per channel, the mean and standard deviation, and percentiles at 25%, 50%, 75%, and 95%, along with the minimum and maximum values.

| Value | Count | Mean | Std | Min | 25% | 50% | 75% | 95% | Max |
|---|---|---|---|---|---|---|---|---|---|
| $AOD_{340}$ | 284 | 0.110 | 0.099 | 0.015 | 0.063 | 0.089 | 0.117 | 0.247 | 0.809 |
| $AOD_{380}$ | 284 | 0.107 | 0.087 | 0.020 | 0.067 | 0.087 | 0.114 | 0.227 | 0.710 |
| $AOD_{440}$ | 344 | 0.091 | 0.067 | 0.019 | 0.058 | 0.076 | 0.098 | 0.185 | 0.579 |
| $AOD_{500}$ | 344 | 0.083 | 0.056 | 0.015 | 0.055 | 0.072 | 0.092 | 0.162 | 0.478 |
| $AOD_{675}$ | 344 | 0.064 | 0.038 | 0.009 | 0.042 | 0.057 | 0.074 | 0.128 | 0.285 |
| $AOD_{870}$ | 344 | 0.056 | 0.030 | 0.007 | 0.037 | 0.050 | 0.067 | 0.114 | 0.198 |
| $AOD_{1020}$ | 344 | 0.051 | 0.028 | 0.006 | 0.033 | 0.045 | 0.063 | 0.105 | 0.173 |
| $AOD_{1640}$ | 344 | 0.039 | 0.025 | 0.000 | 0.023 | 0.034 | 0.050 | 0.087 | 0.156 |
| Ang. Exp. | 344 | 0.716 | 0.371 | 0.032 | 0.426 | 0.638 | 0.979 | 1.397 | 1.782 |

three-year study period, largely due to extended periods when the vessel was not on a mission. Figure 3 shows the temporal evolution of these daily means for $AOD_{440}$ and the Ångström Exponent from July 1, 2021, to June 30, 2024.

The mean AOD values from the full dataset in Table 1 are $0.093 \pm 0.075$, $0.063 \pm 0.041$, and $0.054 \pm 0.031$ at 440, 675, and 870 nm, respectively. These values remain consistent when calculated from daily averages, as shown in Table 2, yielding

$0.091 \pm 0.067$, $0.064 \pm 0.038$, and $0.056 \pm 0.030$ for the same three channels. Additionally, the mean Ångström Exponent is $0.75 \pm 0.40$ for the entire dataset and $0.72 \pm 0.37$ when computed from daily averages. These averages confirm the overall pristine atmospheric conditions in the operational area of the R.V. Marion Dufresne, which are characteristic for the Indian Ocean. These findings are consistent with those reported by Mallet et al. (2018) and other studies conducted in the region (references therein) and generally align with observations from other clean marine environments with minimal continental

influence (Smirnov et al., 2002). This is clearly observed in Figure 3, where daily averaged $AOD_{440}$ values rarely exceed 0.2, with the vast majority remaining below 0.1 (limit of the 75th percentile for both the full dataset and the daily means). It is important to note that the Ångström Exponent exhibits a large dispersion, as reflected in its standard deviation of 0.37 and clearly visible in Figure 3. This variability is partly due to the very low AOD conditions frequently observed in the region. For instance, the mean $AOD_{870}$ is 0.056, with a median of 0.05, while the associated uncertainty in AOD retrievals is approximately

0.01 (Eck et al., 1999). This means that at least 50% of the data in this channel have an uncertainty of at least 20% in the AOD measurement itself, which significantly impacts the accuracy of Ångström Exponent calculations under these low aerosol load conditions and may explain a large part of the observed variation.

Long-term observations from the AERONET Saint-Denis site provide a valuable reference for assessing aerosol conditions in the operational area of the R.V. Marion Dufresne. The data from this site was previously used to contextualize the transition

rates between AOD data quality levels for the R.V. Marion Dufresne in subsection 2.2 and will be further utilized in a point-



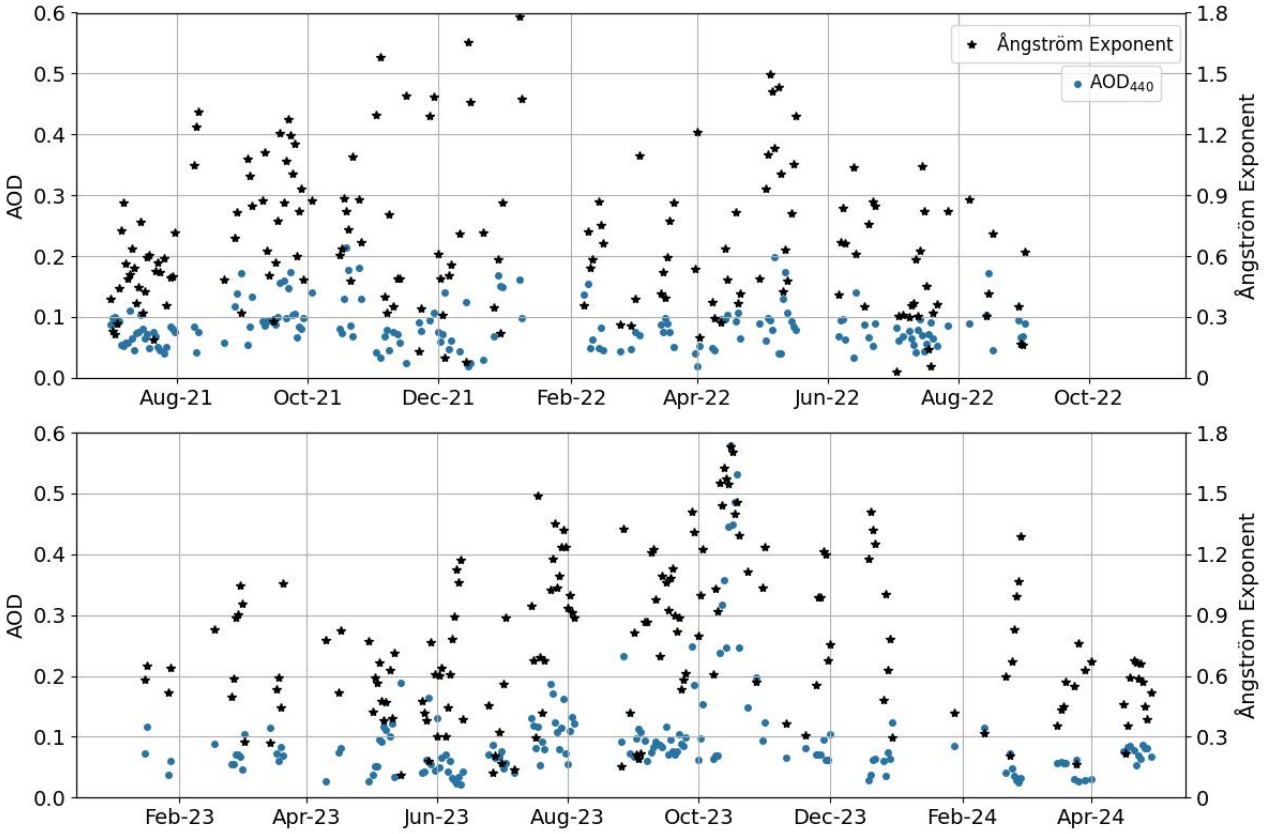

**Figure 3.** Time series of daily mean AOD Level 1.5 data at 440 nm (blue dots) and the Ångström Exponent (black asterisks, right y-axis) recorded by the shipborne-adapted CIMEL CE318-T photometer aboard the R.V. Marion Dufresne from July 1, 2021, to June 30, 2024.

by-point comparison in subsection 3.2. For the period 2007–2019, Duflot et al. (2022) reported a mean $AOD_{440}$ of 0.08 and an Ångström Exponent (440/870) of 0.71, values that closely match those obtained in this study (0.09 and 0.72, respectively, from daily averages). According to Duflot et al. (2022), these averages reflect the regional dominance of marine aerosols, providing a representative baseline for the local atmospheric column. Comparable statistics for the Ångström Exponent can be found in

Smirnov et al. (2011), whose Figure 5c (MAN climatology from Indian Ocean) shows a similar variation range[5] with a 50% percentile of approximately 0.6, slightly lower than the 0.64 observed in this study for the daily averages (Table 2).

---

[5]It is worth noting that while a minimum Ångström Exponent value of -0.823 is reported here, less than 0.6% of the data are negative. Similarly, negative values have also been observed in the region by Smirnov et al. (2011). These occurrences are generally associated with very low AOD conditions (AOD440 < 0.05), where the relative uncertainty of AOD measurements (estimated at 0.01 for standard channels and 0.02 for UV channels; see Eck et al., 1999) can range from 20% to 100% for $AOD_{440}$, considering that the minimum observed values are 0.019 for daily means and 0.01 for the full dataset. This uncertainty is even higher for $AOD_{870}$, where AOD values are generally lower. These large relative errors significantly impact the accuracy of Ångström Exponent calculations.



The study by Duflot et al. (2022) highlights the significant seasonal impact of biomass burning emissions, primarily due to bushfires, particularly during the September–October–November (SON) period. During the extended dry season, which spans from late July to early December (shortly before the onset of the first significant rains), emissions from southeastern Africa and Madagascar lead to a substantial increase in AOD values and Ångström Exponents, indicating a greater dominance of fine-mode particles in the region. Duflot et al. (2022) notes that while sea salt aerosols are consistently present throughout the year and contribute significantly to AOD, they account for only $16.3\%$ of its variability. In contrast, biomass burning emissions dominate this variability, contributing $67.4\%$. Thus, sea salt establishes the regional AOD baseline, while biomass burning plumes drive the most significant fluctuations during the dry season.

An extraordinary biomass burning plume was detected aboard the R.V. Marion Dufresne from October 11 to 20, 2023, in the vicinity of La Réunion. As seen in Figure 3, this period stands out as the only one in the entire dataset where daily mean $AOD_{440}$ values exceed 0.3. Notably, $65\%$ of the data within the top $5\%$ percentile of $AOD_{440}$ for the three-year dataset were recorded during this event. Furthermore, all measurements in the top $2.1\%$ percentile (approximately 580 measurements with $AOD_{440}$ exceeding 0.36) occurred between October 12 and 19, 2023. The highest $AOD_{440}$ value recorded in the dataset (0.73, see Table 1) and the highest daily average (0.58, see Table 2) both occurred on October 16, 2023. The weekly average $AOD_{440}$ and Ångström Exponent reached 0.45 and 1.56, respectively well above the 95th percentile of $AOD_{440}$ (0.21) and Ångström Exponent (1.46) for the three-year dataset (Table 1).

While Duflot et al. (2022) acknowledges the presence of biomass burning events in the region and their impact on AOD, their study does not provide a detailed analysis of individual episodes concerning aerosol optical and microphysical properties. Other studies, such as Clain et al. (2009), Duflot et al. (2010), Vigouroux et al. (2012), and Verreyken et al. (2020), primarily focus on the gas-phase contributions associated with individual biomass burning events in the region, without providing insights into aerosol properties. On the other hand, Smirnov et al. (2011) documented a similar episode in the southwest Indian Ocean region on November 12, 2009, where $AOD_{500}$ measured from a vessel reached approximately 0.60 with an Ångström Exponent of about 1.4. However, this episode was detected in the Mozambique Channel and likely did not extend to La Réunion, as $AOD_{500}$ values measured at the Saint-Denis site (Level 2) for November 2009 rarely exceeded 0.15, except for a few measurements on November 5. Additionally, no detailed aerosol retrievals analyzing properties such as size distribution or optical parameters were performed for this event, as Smirnov et al. (2011) relied on MAN data, which lack the sky radiance measurements required for AERONET aerosol retrievals.

Note that sky radiance observations were collected alongside automated AOD measurements aboard the R.V. Marion Dufresne during that week, enabling the retrieval of microphysical and optical aerosol properties using AERONET's standard retrieval algorithm. The exceptionally high AOD values measured during the biomass burning event of October 11–20, 2023, provide a unique opportunity to evaluate the first quality-assured AERONET aerosol retrievals ($AOD_{440} > 0.4$) obtained from a shipborne platform. A detailed analysis of this event and the retrieved optical and microphysical aerosol properties will be presented in Section 4.



## 3.2 Data validation

As mentioned in the Introduction, an initial validation of AOD data from the shipborne version of the CIMEL CE318-T photometer was carried out by Yin et al. (2019), who compared automated AOD measurements with a MAN Microtops II instrument aboard the R.V. Polarstern during the OCEANET transatlantic campaigns (PS113, PS116, and MOSAIC/Arctic). However, a similar comparison could not be performed in this study, as no Microtops measurements were conducted aboard the R.V. Marion Dufresne during the three-year period from July 2021 to June 2024.

During the study period, two different photometers were used for measurements. The first instrument, labeled #1273, operated from July 1, 2021, until June 12, 2023, when it was scheduled for calibration and replacement. The timing of this change coincided with the Amaryllis-Amagas campaign and the Transama transit campaign, during which the vessel was relocated from its regular operational area in the southwestern Indian Ocean to the Brazilian coast (more information about these campaigns is available at https://www.ipsl.fr/campagne/amaryllis-amagas/ and https://archimer.ifremer.fr/doc/00875/98738/108483.pdf, last accessed: December 31, 2024). During Transama campaign, both the outgoing instrument (#1273) and its replacement (#1243) were kept onboard, enabling simultaneous operation between April and June 2023. This overlap allowed for additional tests and optimizations[6] conducted by research engineers participating in the campaign. For the time series presented in the previous subsection, data from instrument #1273 were used until May 31, 2023, after which data from instrument #1243 were utilized until June 30, 2024.

Figure 4 presents the comparison of coincident data, using Level 1.5 AOD from both instruments (the highest quality level achievable for moving platform data). This includes AOD measurements at (from top to bottom and left to right) 380, 440, 500, 675, and 870 nm, as well as the Ångström Exponent. In this analysis, AOD measurements from photometer #1273 (plotted on the x-axis) were compared with the average AOD values from photometer #1243 (plotted on the y-axis) within a $\pm 3$ minute interval. All AOD datasets include 1180 data points, except for the 380 nm channel which is limited to 622 points since $AOD_{380}$ is provided only during daytime (Moon-based AOD measurements do not include this wavelength, as explained in subsection 2.2). The Ångström Exponent $440/870$ is also based on 1180 points. Color bars represent data density, using a $0.01 \times 0.01$ grid for AOD comparisons and a $0.1 \times 0.1$ grid for the Ångström Exponent. A logarithmic scale was applied to the density representation to better visualize the high concentration of data points along the one-to-one line. The agreement between the two instruments is particularly strong for AOD comparisons, with $R > 0.96$ for all AOD channels and slopes ranging from 0.92 to 0.98.

Regarding AOD comparisons, RMSE values range from 0.005 to 0.008, which are slightly lower than those reported by Yin et al. (2019), where differences relative to the Microtops-II instrument ranged from 0.009 to 0.015 (see Figure 4 in their study). The biases in AOD measurements are also minimal, with values of $4.2 \times 10^{-3}$, $-1.7 \times 10^{-3}$, $-8.0 \times 10^{-5}$, $1.6 \times 10^{-3}$, and $2.0 \times 10^{-3}$ for the 380, 440, 500, 675, and 870 nm channels, respectively. These bias values, defined as AOD(#1273)

---

[6]For instance, a shorter collimator was tested to reduce wind-induced instabilities, as wind significantly affects data collection at sea. Additionally, software modifications were implemented, and the two photometers were placed in different locations on the vessel to evaluate the effects of ship vibrations and movements. Elastic lidar measurements were also performed, enabling Klett inversions based on total extinction measurements derived from the photometer and lidar signals. Further details on the Amaryllis-Amagas/Transama campaign can be found in Sanchez-Barrero (2024); Sanchez-Barrero et al. (2025)



- AOD(#1243), are slightly smaller than those reported in Yin et al. (2019), where they ranged from $2 \times 10^{-3}$ to $5 \times 10^{-3}$. Notably, the high correlations and low biases observed for AOD comparison in this study are comparable to those obtained in ground-based calibration sites during intercalibration periods with AERONET master instruments, demonstrating the reliability of shipborne Level 1.5 data.

485 However, the comparison for the Ångström Exponent shows a much higher RMSE of $0.158$ compared to $0.063$ observed in Yin et al. (2019). The correlation coefficient here is R=0.73, which is lower than the R=0.9 obtained by Yin et al. (2019). These larger discrepancies in the Ångström Exponent, despite smaller differences in AOD, might be attributed to the significantly low AOD values observed during the Amaryllis-Amagas/Transama campaign, which took place under pristine conditions (with $AOD_{440}$ below $0.18$, except for one night with measurements reaching up to $0.27$), compared to the higher AOD values

490 observed in the analysis by Yin et al. (2019) (with values reaching $0.6$ for $AOD_{440}$). As mentioned in the previous section, when AOD values are low, the relative errors tend to be much larger, which has a more pronounced impact on the Ångström Exponent. For instance, if the analysis is restricted to cases where $AOD_{870} > 0.05$ (not shown in any figure, with 490 data points), the RMSE for the Ångström Exponent decreases to $0.12$, and the correlation coefficient increases to $0.8$.

 An additional validation of the shipborne CIMEL CE318-T is conducted using the AERONET ground-based photometer

495 at the Saint-Denis site, previously introduced in subsection 3.1. Located 93 meters above sea level and approximately 20 kilometers from the port of La Réunion, this site offers a unique opportunity for comparison due to the frequent proximity of the R.V. Marion Dufresne during its regular entries and departures. As the nearest ground-based AERONET site with the highest number of coincident measurements, it serves as a key validation point for the shipborne data, despite the inherent challenges posed by the elevation difference.

500 Figure 5 presents the correlation between AOD and Ångström Exponent data from the CIMEL CE318-T at the Saint-Denis AERONET site and the shipborne photometer aboard the R.V. Marion Dufresne. These comparisons were conducted over the full three-year period of data collection. To ensure meaningful comparisons, measurements were selected when the R.V. Marion Dufresne was within a maximum distance of 50 kilometers from the Saint-Denis site. Notably, adjusting the threshold between 30 and 100 km had only a minor impact on the dataset size, with 1745 points at 30 km, 1833 points at 50 km (the selected

505 threshold), and 1941 points at 100 km. The statistical results (e.g., RMSE, slopes, and correlation coefficients) remained largely unchanged across these thresholds. However, a significant reduction occurs when using a stricter 20 km limit, leaving only 8 coincident observations. This sharp drop is explained by the fact that the distance between the ship's docking location in Le Port and the Saint-Denis photometer is approximately 21 km, meaning that most entries and departures fall just outside the 20 km range.

510 For each comparison, the shipborne data were averaged over a $\pm 3$-minute interval relative to the ground-based measurements. The figure follows the same structure as Figure 4, displaying AOD comparisons at 380, 440, 500, 675, and 870 nm, along with the Ångström Exponent (arranged top to bottom, left to right). Under these conditions, 1834 coincident observations were identified. To ensure the highest data quality, only Level 2 AERONET data from Saint-Denis were used in this comparison. Since Moon-based AERONET observations are still under evaluation and have not yet reached Level 2 status, the

515 comparison includes only AOD values obtained from direct Sun measurements.





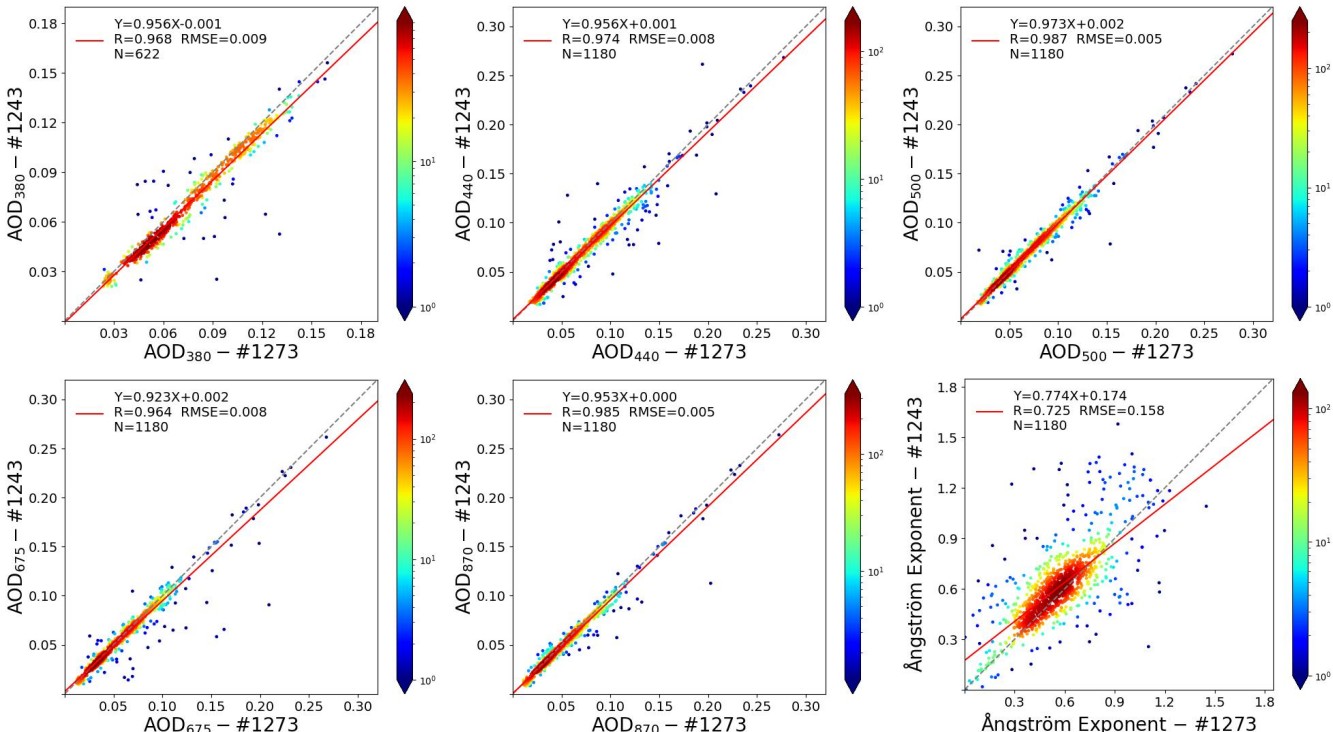

**Figure 4.** Comparison of Level 1.5 AOD and Ångström Exponent measurements between photometer #1273 (x-axis) and photometer #1243 (y-axis) during the Amaryllis-Amagas/Transama campaign from late April to early June 2023. The comparison is shown for the AOD channels 380, 440, 500, 675, and 870 nm (top to bottom, left to right) and the Ångström Exponent. Color bars represent data density on a $0.01 \times 0.01$ grid for AOD and a $0.1 \times 0.1$ grid for the Ångström Exponent, with a logarithmic scale applied due to the high data density along the one-to-one line. For each AOD measurement from photometer #1273, the comparison was made with the average of AOD data from photometer #1243 within a $\pm 3$-minute interval. All data shown have undergone the Level 1.5 cloud-screening and quality control procedures described in subsection 2.2 and references therein.

The correlations between the shipborne and ground-based photometers are generally strong, with $R$ values ranging from 0.86 to 0.93. However, a clear negative bias is observed when using the Saint-Denis photometer as a reference. The bias values are $-1.3 \times 10^{-2}$, $-9.5 \times 10^{-3}$, $-8.7 \times 10^{-3}$, $-8.8 \times 10^{-3}$, and $-7.0 \times 10^{-3}$ for the 380, 440, 500, 675, and 870 nm chan-
nels, respectively. These biases are approximately an order of magnitude larger than those found in the previous instrument intercomparison, likely due to the altitude difference between the Saint-Denis station (93 m above sea level) and sea level. Additionally, notable differences in local pollution levels may contribute to these discrepancies. While Saint-Denis is a relatively cleaner urban site, the R.V. Marion Dufresne is often docked in Le Port, an area heavily influenced by industrial emissions, including those from a coal-fired power plant.

Upon examining the RMSE values between the shipborne and ground-based photometers, it is noted that they are approxi-
mately double the typical AERONET error estimates for the respective wavelengths. Specifically, the RMSE values are 0.037,





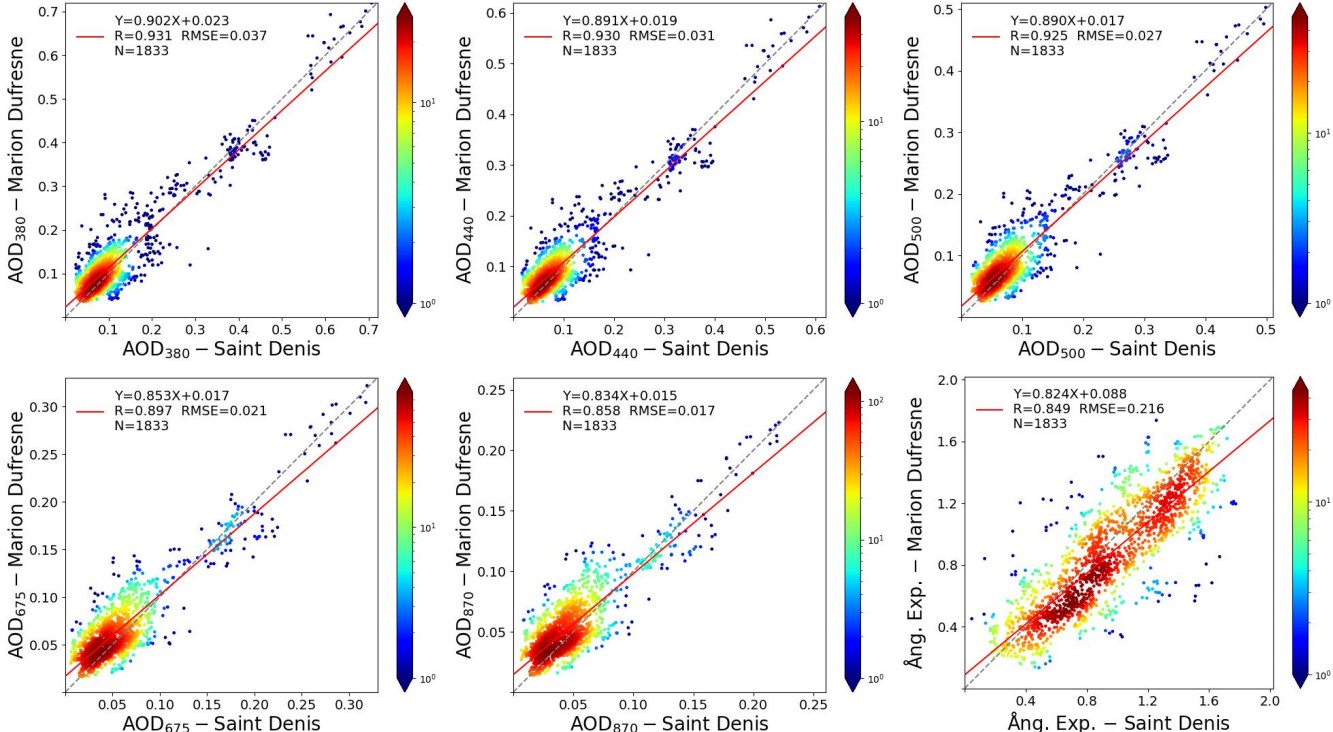

**Figure 5.** Comparison of AOD and Ångström Exponent measurements between the ground-based photometer at Saint-Denis (x-axis) and the shipborne photometer aboard the R.V. Marion Dufresne (y-axis). The comparison is shown for the AOD channels 380, 440, 500, 675, and 870 nm (top to bottom, left to right) and the Ångström Exponent. Color bars represent data density on a $0.01 \times 0.01$ grid for AOD and a $0.1 \times 0.1$ grid for the Ångström Exponent, with a logarithmic scale used due to strong data density along the one-to-one line. For each AOD measurement from the R.V. Marion Dufresne, the comparison was made with the average of AOD data from the Saint-Denis photometer within a $\pm 3$-minute interval, and only when the ship was within a maximum distance of 50 km from the Saint-Denis site.

0.031, 0.027, 0.021, and 0.017 for the 380, 440, 500, 675, and 870 nm channels, respectively (AERONET error estimates are around 0.02 for the 380 nm channel and 0.01 for the channels 675 and 870 nm, for 440 and 500 nm an intermediate value of 0.015 can be expected Eck et al., 1999). Despite these higher RMSE values, the agreement between the shipborne and ground-based measurements remains within acceptable limits. Even if the altitude difference were not considered, the comparison between the two instruments would still be reasonable.

There is also a positive bias for the Ångström Exponent of 0.08. The larger values at the Saint-Denis site could be attributed to a diminished influence of marine aerosols and a relatively greater contribution from local urban aerosols in the city, primarily consisting of smaller particles, as reported by Duflot et al. (2022). Nevertheless, a relatively strong correlation for the Ångström Exponent is observed, better than that seen in Figure 4, indicating that both instruments are identifying similar aerosol types in the atmospheric column, with the primary difference being the lowest 93 meters, which may lead to the observed bias in AOD and Ångström Exponent measurements. The greater variability of the Ångström Exponent during the three years analyzed



in this study, compared to the Amaryllis-Amagas/Transama campaign, also positively contributes to the improved correlation coefficient.

## 4 First quality-assured AERONET aerosol retrievals from a boat

The use of the modified CIMEL CE318-T photometer for shipborne platforms has enabled the acquisition of sky radiance measurements, a necessary input for performing aerosol retrievals with the AERONET aerosol algorithm. As discussed in Section 2.2, this algorithm retrieves detailed binned aerosol volume size distributions and spectrally independent optical properties, including refractive index as a primary parameter and derived properties such as single scattering albedo and absorption (Dubovik and King, 2000). The systematic collection of sky radiances over the past three years as part of the MAP-IO program

aboard the R.V. Marion Dufresne represents a significant milestone, marking the first AERONET aerosol inversions conducted on a shipborne platform.

Among the potential retrievals available from the three-year dataset, those from the week of October 12 to October 19, 2023, stand out as particularly noteworthy. As discussed in Section 3.1, this period corresponds to an extraordinary biomass burning event linked to intense bushfires in Madagascar, which significantly elevated aerosol levels in the surrounding region.

Madagascar's bushfire season, which typically spans the dry season from late July to early December (Clain et al., 2009; Duflot et al., 2010, 2022; Vigouroux et al., 2012; Verreyken et al., 2020) as in the rest of southern Africa area, reached a peak in mid-October 2023. During this time, fire alerts were exceptionally high, with the most intense activity concentrated in the western regions of Madagascar. According to the Global Disaster Alert and Coordination System (GDACS), approximately 5,154 hectares were burned during this period. The rigorous criteria required for AERONET Level 2 aerosol optical inversion

properties[7] —$AOD_{440} > 0.4$ and $\theta_S > 50°$ (Dubovik et al., 2000; Holben et al., 2006)— were met several times during this period, allowing for the first detailed analysis of retrievals from shipborne measurements under sufficient aerosol load. The results of these retrievals are presented in this section.

The top-left panel of Figure 6 illustrates the aerosol and fire activity around La Réunion during this exceptional week. This composite image, obtained from NASA Worldview, overlays a MODIS RGB image with the AOD product at 550 nm from

October 16, derived using the C6.1 MODIS Dark Target (DT) algorithm. The DT algorithm, akin to the Collection 6 version described by Levy et al. (2013), incorporates updates such as enhanced sensor calibration, stricter cloud screening, and an improved surface reflectance model for urban areas (Gupta et al., 2016). The fire counts layer, based on the detection algorithm by Giglio et al. (2003) and depicted as red spots, underscores the widespread distribution of fires across Madagascar during this period. The position of the R.V. Marion Dufresne on October 16, 2023, at 12:11 UTC, is marked with a black cross (20.93°S,

55.20°E) in the figure. This moment represents the first almucantar measurement meeting Level 2 aerosol retrieval criteria[8], not

---

[7]It is worth noting that the Level 2 criteria for size distribution properties are less stringent, as they do not impose an $AOD_{440}$ threshold. However, they share the remaining conditions with the optical property criteria, such as $\theta_S > 50°$ and a minimum symmetrical number of radiance measurements in the almucantar (or, more recently, in hybrid scans) across different scattering angle ranges, among other. For further details, refer to Table 3 in Holben et al. (2006).

[8]In this study, retrievals meeting the thresholds of $AOD_{440} > 0.4$ and $\theta_S > 50°$ are referred to as Level 2 to distinguish them from those that do not meet these thresholds. However, official Level 2 status requires that the underlying AOD data be validated and reclassified to Level 2 within the AERONET





**Figure 6.** Overview of the aerosol and fire conditions around La Réunion during the week of October 12–19, 2023. Top-left panel: Composite image obtained from NASA Worldview, combining MODIS RGB imagery with the $AOD_{550}$ from October 16 (C6.1 MODIS Dark Target algorithm), overlaid with fire counts detection. The black cross marks the position of the R.V. Marion Dufresne on October 16 (20.93 E, 55.20 S), approximately 12 km northwest of the closest coastal point of La Réunion and 30 km from Saint-Denis. Top-right panel: Backward air mass trajectories for October 16, calculated using the HYSPLIT model. Bottom-left panel: AOD values recorded during the test in the sea area campaign (September 29 to October 30, 2023). Bottom-right panel: Time series of hourly-averaged spectral AOD (Level 1.5) values from October 10 to October 21, represented by navy blue (340 nm), violet (380 nm), light blue (440 nm), green (500 nm), yellow (675 nm), orange (870 nm), red (1020 nm), and brown (1640 nm) dots. The Ångström Exponent is shown with black asterisks (right y-axis).

only during this event but also across the entire dataset recorded aboard the R.V. Marion Dufresne since 2021. At this location,

network. Shipborne measurements are currently classified as Level 1.5, and retrievals satisfying these additional criteria will only be officially recognized as Level 2 once the AOD data are upgraded. Retrievals failing to meet these thresholds will remain classified as Level 1.5 even after reclassification.



the vessel was approximately 12 km northwest of the nearest coastal point of La Réunion and 30 km from the AERONET station in Saint-Denis. The top-right panel of Figure 6 displays back-trajectories, calculated using the HYSPLIT model (Stein et al., 2015), that estimate the origins of the air masses over the R.V. Marion Dufresne on October 16, 2023, at 12:00 UTC, around the time of the almucantar measurement from the boat. At 3000 meters, the air mass forms a loop around Madagascar, traveling counterclockwise from the southwest to the northwest before exiting to the east. At 2000 meters, the air mass crosses the southern part of the island, while at 1000 meters, it remains localized near La Réunion, indicating recirculation and certainly more affected by marine aerosols. These trajectories, calculated over a five-day period with a point every six hours, confirm that the air masses influencing the aerosol observations over the ship predominantly originated from Madagascar, where the biomass burning episode occurred.

The bottom-left panel of Figure 6 illustrates the $AOD_{500}$ recorded between September 29 and October 30, 2023, during the campaign referred to as test in the sea area. These geolocated AOD measurements reveal the ship's trajectory, showing a voyage to northern Madagascar at the beginning of October, characterized by typically pristine AOD values, and shorter trips of two to three days around the port of Saint-Denis from mid-October until the end of the month. Some of these shorter trips coincided with the biomass burning episode. These AOD data are available on the PHOTONS system platform (https://mobile.photons.univ-lille.fr/, last access: 31 December 2024), and the figure can be directly generated from the web by selecting the R.V. Marion Dufresne and the campaign "2023 MAP-IO Test en mer 2023-10."

The bottom-right panel presents multi-spectral Level 1.5 AOD data recorded by the shipborne-adapted CIMEL318-T photometer from October 10 to October 21, covering the week of interest. The values are displayed as hourly averages for all standard AERONET channels: 340 nm (navy blue dots), 380 nm (violet dots), 440 nm (light blue dots), 500 nm (green dots), 675 nm (yellow dots), 870 nm (orange dots), 1020 nm (red dots), and 1640 nm (brown dots). The temporal evolution of the Ångström Exponent is represented by black asterisks (right y-axis and also displayed as hourly averages). On October 10, conditions reflect the typical pristine marine environment, dominated by sea salt aerosols, observed during the three-year analysis period. The mean $AOD_{440}$ for this day was 0.076, with an Ångström Exponent of 0.88—values consistent with the dataset's overall mean. However, due to abundant cloud cover during this period (see Figure 6), no data were recorded in the afternoon on October 10, and only limited measurements were obtained on October 11, primarily in the late afternoon. Starting on October 11, a gradual increase in AOD values, indicative of rising aerosol load, and Ångström Exponent values, reflecting fine-mode particle dominance, becomes evident. Between October 12 and October 19, $AOD_{440}$ values consistently exceeded 0.36, corresponding to the top 2.1% percentile of the three-year dataset (approximately 580 observations, as noted in subsection 3.1). Weekly averages for this period were 0.45 for $AOD_{440}$ and 1.56 for Ångström Exponent, the latter significantly exceeding the 95% percentile value of 1.46. The highest AOD levels were recorded on October 16, with a maximum $AOD_{440}$ value of 0.73. This day, highlighted in the MODIS composite (top-left panel of Figure 6), recorded a MODIS $AOD_{550}$ of 0.52, closely aligning with the shipborne $AOD_{500}$ values, which averaged $0.49 \pm 0.05$ and peaked around midday. After October 19, AOD values gradually declined, with limited measurements on October 20 reflecting this reduction. Persistent cloud cover prevented further AOD data collection until October 31, except for a few observations during the night of October 24. By October 31, atmospheric conditions had returned to a typical clean marine state, with a daily average $AOD_{440}$ of 0.096 and an Ångström





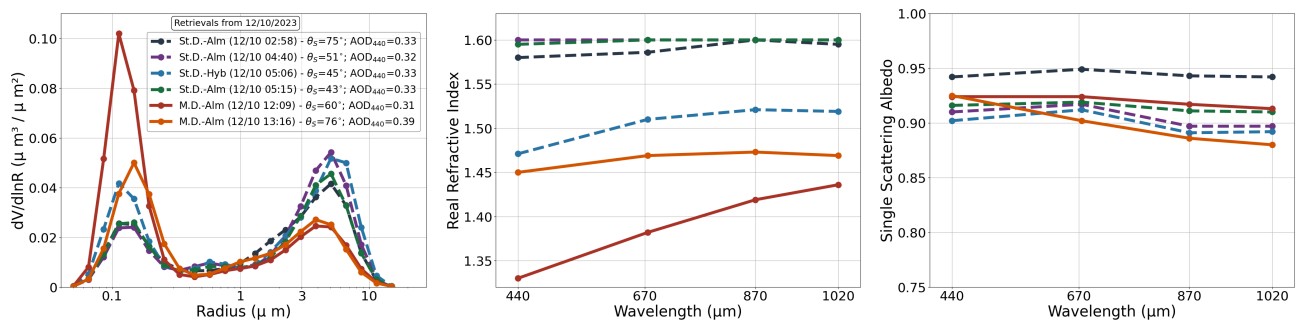

**Figure 7.** Aerosol retrievals performed on October 12, 2023, at the R.V. Marion Dufresne (solid lines) and Saint-Denis (dashed lines). Panels, from left to right, represent the retrieved 22 bin volume size distribution, the real part of the refractive index, and the single scattering albedo at 440, 675, 870 and 1020 nm. The inversions include morning measurements from Saint-Denis and afternoon measurements from the R.V. Marion Dufresne. The legend specifies the type of scan, exact time of radiance measurement, the solar zenith angle and AOD$_{440}$.

Exponent of 1.02, consistent with baseline marine aerosol conditions. It should be noted that the marked days in the graph correspond to 00:00 UTC. Due to the geographic location and time of the year, the first morning direct Sun measurements (daylight AOD) began around 2:30 a.m. UTC, provided cloud conditions allowed. Additionally, the new moon occurred on the 

night of October 14 to October 15, 2023, preventing moon-based AOD measurements during the period from October 8 to 21 (from the last quarter to the first quarter of the lunar phase).

Figures 7 and 8 present the aerosol inversions performed at the R.V. Marion Dufresne and Saint-Denis sites during the week of October 12–19, 2023. The inversions from October 12 are shown separately in Figure 7, as it is the only day with simultaneous measurements from both sites. Figure 8 includes the remaining inversions, consisting of data from the R.V. Marion 

Dufresne on October 16 and 17, and from Saint-Denis site on October 19. Due to persistent cloud cover during this period, complete inversions were only obtained at Saint-Denis on October 12 and 19, and aboard the R.V. Marion Dufresne on October 12, 16, and 17. In each figure, the panels from left to right represent the volume size distribution, the real part of the refractive index, and the single scattering albedo. The retrievals from the R.V. Marion Dufresne are represented by solid lines, while those from Saint-Denis are shown as dashed lines. The legend in each figure includes the type of scan, the exact date and time 

of the radiance measurement corresponding to the inversion, the solar zenith angle $\theta_S$ and the AOD$_{440}$.

Additionally, Table 3 provides a summary of the key parameters derived from the inversions during this period. The first columns of the table include information similar to the figure legends: site name, date, type of scan, solar zenith angle ($\theta_S$), Ångström Exponent ($\alpha$), and AOD$_{440}$. The table further presents a concise overview of the aerosol size distribution parameters for both fine and coarse modes, including modal radius (R$_{Vf}$ and R$_{Vc}$ in [$\mu$m]) and volume concentration (C$_{Vf}$ and C$_{Vc}$ in 

[$\mu$m$^3$/$\mu$m$^2$]). The total effective radius (R$_{eff}$ in [$\mu$m]) is also included. It is important to highlight that these size distribution parameters are derived systematically from the detailed 22-bin size distribution provided by the retrieval algorithm, rather than being direct outputs. Additional details on the methodology to obtain these size parameters can be found in the AERONET in-



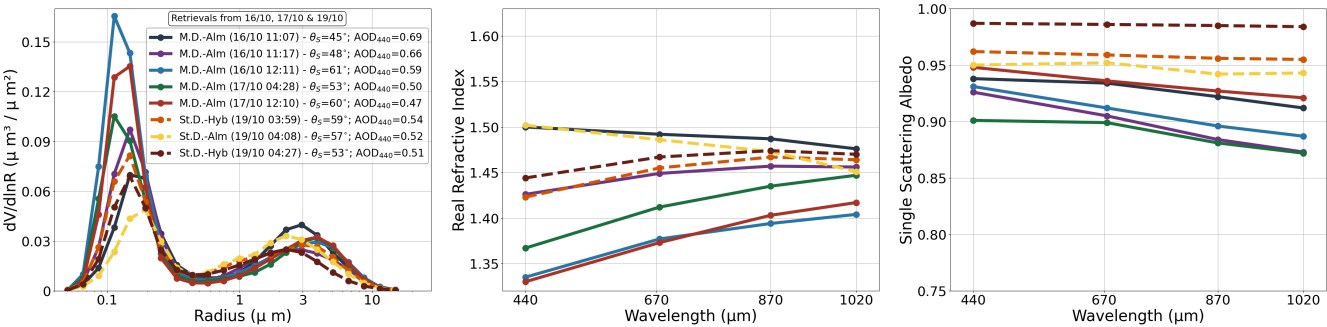

**Figure 8.** Aerosol retrievals performed at the R.V. Marion Dufresne (October 16 and 17) and Saint-Denis (October 19), 2023. Panels, from left to right, display the retrieved 22 bin volume size distribution, the real part of the refractive index, and the single scattering albedo at 440, 675, 870 and 1020 nm. Retrievals from the R.V. Marion Dufresne are shown as solid lines, while those from Saint-Denis are represented as dashed lines. The legend indicates the type of scan, measurement time, solar zenith angle and $AOD_{440}$.

version product documentation (http://aeronet.gsfc.nasa.gov/new_web/Documents/Inversion_products_V2.pdf, last accessed: 31 December 2024). Finally, the table summarizes key aerosol optical properties, such as the real part of the refractive index and the single scattering albedo (SSA) at 440 nm, offering a comprehensive view of some optical aerosol properties observed during this period.

The inversions from October 12, shown in Figure 7, include morning measurements from the Saint-Denis site and afternoon measurements from the R.V. Marion Dufresne. Due to cloud cover, no valid inversions were obtained from Saint-Denis in the afternoon or from the R.V. Marion Dufresne in the morning. During this day, the R.V. Marion Dufresne was located in the port area of La Réunion, approximately 30 km from the Saint-Denis site. $AOD_{440}$ values remained relatively constant throughout the day, approximately 0.32–0.33, except for the last inversion from the R.V. Marion Dufresne, which recorded a higher value of 0.39. This evolution is also reflected in the R.V. Marion Dufresne measurements presented in Figure 6. The Ångström Exponent shows a more noticeable evolution, increasing from 1.2–1.3 in the morning to 1.5–1.6 in the afternoon (values presented in Table 3 individually for each retrieval). This shift suggests a greater dominance of fine-mode aerosols (biomass burning) over coarse-mode aerosols (pristine marine aerosol) as the day progresses. This evolution in the Ångström Exponent could explain why coarse-mode aerosols are more prominent in the Saint-Denis inversions compared to those from the R.V. Marion Dufresne in the left panel of Figure 7. This trend is further supported by the values of $R_{eff}$ presented in Table 3, where the morning inversions from Saint-Denis range between 0.37–0.45 $\mu$m, while the afternoon inversions from the R.V. Marion Dufresne fall between 0.18–0.25$\mu$m.

Regarding fine-mode aerosols, while the modal median radius ($R_{Vf}$) remains consistent across inversions from both sites throughout the day (ranging from 0.13 to 0.16 $\mu$m), significant differences are observed in concentration values. Notably, higher fine-mode volume concentration values ($C_{Vf}$) are retrieved from the shipborne data (top-right panel, ranging from 0.047 to 0.079 $\mu$m$^3$/$\mu$m$^2$) compared to those obtained at Saint-Denis (ranging from 0.026 to 0.044 $\mu$m$^3$/$\mu$m$^2$). Additionally,



**Table 3.** Summary of aerosol inversion parameters obtained at the R.V. Marion Dufresne and Saint-Denis sites during the week of October 12–19, 2023. The table includes the site, date, scan type (almucantar or hybrid), solar zenith angle ($\theta_S$), Ångström Exponent ($\alpha$), AOD$_{440}$, size distribution parameters for fine and coarse modes (modal radius $R_{Vf}$ and $R_{Vc}$ in [$\mu$m] and volume concentration $C_{Vf}$ and $C_{Vc}$ in [$\mu m^3/\mu m^2$]), the total effective radius ($R_{eff}$ in [$\mu$m]), and optical properties including the real refractive index and single scattering albedo at 440 nm..

| Site | Date | SkyScan | $\theta_S$ | AOD$_{440}$ | $\alpha$ | R$_{Vf}$ | C$_{Vf}$ | R$_{Vc}$ | C$_{Vc}$ | R$_{eff}$ | RRI$_{440}$ | SSA$_{440}$ |
|------|------|---------|-----------|-------------|----------|----------|----------|----------|----------|-----------|-------------|-------------|
| St.Denis | 12/10 02:58 | Alm | 75° | 0.33 | 1.40 | 0.15 | 0.028 | 3.11 | 0.065 | 0.40 | 1.58 | 0.94 |
| St.Denis | 12/10 04:40 | Alm | 51° | 0.32 | 1.25 | 0.15 | 0.026 | 3.30 | 0.075 | 0.45 | 1.60 | 0.91 |
| St.Denis | 12/10 05:06 | Hyb | 45° | 0.33 | 1.32 | 0.14 | 0.044 | 4.33 | 0.070 | 0.37 | 1.47 | 0.90 |
| St.Denis | 12/10 05:15 | Alm | 43° | 0.33 | 1.34 | 0.15 | 0.028 | 3.22 | 0.063 | 0.41 | 1.59 | 0.92 |
| Mar.Dufr. | 12/10 12:09 | Alm | 60° | 0.31 | 1.52 | 0.13 | 0.079 | 2.91 | 0.041 | 0.18 | 1.33 | 0.92 |
| Mar.Dufr. | 12/10 13:16 | Alm | 76° | 0.39 | 1.65 | 0.16 | 0.047 | 2.71 | 0.045 | 0.26 | 1.45 | 0.92 |
| Mar.Dufr. | 16/10 11:07 | Alm | 45° | 0.69 | 1.74 | 0.18 | 0.070 | 2.64 | 0.061 | 0.28 | 1.50 | 0.94 |
| Mar.Dufr. | 16/10 11:17 | Alm | 48° | 0.66 | 1.73 | 0.16 | 0.089 | 2.50 | 0.049 | 0.22 | 1.43 | 0.93 |
| Mar.Dufr. | 16/10 12:11 | Alm | 61° | 0.59 | 1.72 | 0.14 | 0.139 | 2.81 | 0.052 | 0.17 | 1.33 | 0.93 |
| Mar.Dufr. | 17/10 04:28 | Alm | 53° | 0.50 | 1.65 | 0.14 | 0.094 | 2.76 | 0.051 | 0.19 | 1.37 | 0.90 |
| Mar.Dufr. | 17/10 12:10 | Alm | 60° | 0.47 | 1.65 | 0.14 | 0.112 | 2.90 | 0.052 | 0.19 | 1.33 | 0.95 |
| St.Denis | 19/10 03:59 | Hyb | 59° | 0.54 | 1.49 | 0.15 | 0.071 | 2.20 | 0.058 | 0.24 | 1.42 | 0.96 |
| St.Denis | 19/10 04:08 | Alm | 57° | 0.52 | 1.51 | 0.18 | 0.048 | 2.08 | 0.063 | 0.34 | 1.50 | 0.95 |
| St.Denis | 19/10 04:27 | Hyb | 53° | 0.51 | 1.53 | 0.16 | 0.066 | 1.95 | 0.048 | 0.24 | 1.44 | 0.99 |

lower real refractive index (RRI) values are retrieved from the R.V. Marion Dufresne inversions compared to Saint-Denis retrievals. The Saint-Denis inversions at 02:58, 04:40, and 05:15 UTC exhibit $C_{Vf}$ values that are likely underestimated, accompanied by saturated RRI values reaching the upper limit of 1.6. Conversely, the R.V. Marion Dufresne inversion at 12:09 UTC reveals exceptionally high $C_{Vf}$ values and an RRI nearing the lower limit of 1.33 of AERONET inversion for the 440 nm channel. These results are likely due to the well-documented anti-correlation often observed between RRI and fine-mode size distribution, particularly in fine-mode concentration. This phenomenon is extensively described in the literature (Dubovik et al., 2000, 2002b; Torres et al., 2014; Fedarenka et al., 2016; Sinyuk et al., 2020; Herrera et al., 2022). For instance, Figure 33 of Herrera et al. (2022) illustrates the correlation matrix for retrieval unknowns in a simulated biomass burning scenario, revealing a strong negative correlation (dark blue) between RRI and fine-mode size distribution concentrations. The absence of angular measurements in some scenarios, particularly between 40° and 90° of scattering angle due to cloud screening criteria in almucantar and hybrid scans based on symmetry tests, likely amplifies this anti-correlation in the dataset from both sites during that week. It is worth noting that polarimetric observations can mitigate this anti-correlation and improve retrieval accuracy, as shown in sensitivity studies by Fedarenka et al. (2016). However, neither the shipborne nor the Saint-Denis photometers were equipped with calibrated polarized measurement, limiting further exploration of this issue.



Two inversions—Saint-Denis at 05:06 UTC and R.V. Marion Dufresne at 13:16 UTC—show intermediate and similar values for both the size distribution and RRI, making them the most realistic representations of the columnar aerosol properties for this day. These intermediate RRI values, ranging from 1.45 to 1.52, are reasonably expected for a mix of biomass burning and sea salt aerosols. While Dubovik et al. (2002b) reported an RRI of approximately 1.51 for pure biomass burning aerosols in southern Africa, the mixture with sea salt could reduce the RRI since the value of this aerosol component is typically between 1.35 and 1.40, depending on water uptake of sea salt under high relative humidity conditions (Schuster et al., 2009). It is important to highlight that there are no detailed studies available for direct comparison of the observed columnar RRI values in this specific region, including the south-eastern African Indian Ocean and the Madagascar-La Réunion area, where, under high aerosol load conditions, biomass burning is typically mixed with sea salt aerosols (Duflot et al., 2022). While many studies have focused on the gas-phase contributions associated with biomass burning episodes in the region (Clain et al., 2009; Duflot et al., 2010; Vigouroux et al., 2012; Verreyken et al., 2020), and Duflot et al. (2022) has presented some analysis of aerosol size properties, detailed investigations into the columnar optical properties of aerosols remain scarce, primarily due to the limited number of Level 2 aerosol optical property inversions available for this region. At the Saint-Denis site, for instance, only nine almucantar inversions meet the Level 2 criteria of $AOD_{440} > 0.4$ and $\theta_S > 50°$ across the entire dataset (2003–2024). These inversions are exclusively from the dry season and likely represent a mixture of biomass burning and sea salt aerosols (Duflot et al., 2022). The average RRI values for these nine inversions at Saint-Denis are $1.51 \pm 0.06$, $1.49 \pm 0.05$, $1.48 \pm 0.05$, and $1.46 \pm 0.05$ at 440, 670, 870, and 1020 nm, respectively, and are similar to the intermediate values found in the two aforementioned inversions. The relatively high variability (standard deviation) in these RRI values is likely driven primarily by varying contributions of sea salt aerosols in the mixture, which can lower the RRI compared to that of pure biomass burning aerosols. Another potential contributor to this variability is the previously mentioned anti-correlation, which introduces a relatively high uncertainty in the RRI values derived from AERONET inversions. For reference, Sinyuk et al. (2020) reported an uncertainty of approximately 0.02 in RRI for pure African biomass burning aerosols when $AOD_{440} = 0.4$.

As noted by Sinyuk et al. (2020), despite the anti-correlation between RRI and fine-mode aerosol volume concentration, their combined effects counterbalance in terms of aerosol scattering. Thus, the capability of AERONET-type inversions to accurately distinguish between absorption and scattering remains unaffected, even in cases where the anti-correlation is significant. Consequently, even in scenarios with high uncertainties in RRI and fine-mode volume concentration, the retrieval of SSA is highly accurate under the conditions established by Level 2 retrieval criteria. The SSA values observed in the shipborne and Saint-Denis inversions (right panel) are therefore reliable and show spectral values ranging from 0.88 to 0.95, with most falling between 0.90 and 0.92. These values are less absorbing than the averages typically found in pure biomass burning episodes in southern Africa (e.g., Dubovik et al. (2002a); Giles et al. (2012); Denjean et al. (2020)), which range between 0.75 and 0.85. The higher SSA values observed here are consistent with what is expected for a case of mixed biomass burning and sea salt aerosols—a highly scattering and non-absorbing aerosol type—or, more generally, even for aging biomass burning aerosols influenced by high relative humidity (Mallet et al., 2019). Indeed, the spectral SSA averages of the aforementioned nine AERONET Level 2 inversions from Saint Denis are $0.92 \pm 0.03$, $0.89 \pm 0.03$, $0.87 \pm 0.04$, and $0.85 \pm 0.04$ at 440, 670, 870, and 1020 nm, respectively. The SSA values retrieved from the inversions presented in Figure 7 are broadly consistent with





these averages, though they appear slightly less absorbing overall. Among them, the inversion from the R.V. Marion Dufresne at 13:16 UTC stands out as the closest match to the Level 2 averages at Saint-Denis site, with spectral SSA values of 0.92, 0.90, 0.89, and 0.88 at 440, 670, 870, and 1020 nm, respectively. This inversion has an $AOD_{440}$ of 0.39 and a solar zenith angle of $75.8°$ which is the only one on October 12 that nearly satisfies Level 2 criteria. This retrieval, with an $AOD_{440}$ of 0.39 and a solar zenith angle of $75.8°$, is the only one on October 12 that nearly satisfies Level 2 criteria, further supporting the reliability of shipborne retrievals under conditions of sufficient aerosol load.

Beyond the retrievals performed on October 12, additional inversions were obtained from the R.V. Marion Dufresne on October 16 and 17, and from the Saint-Denis site on October 19, as shown in Figure 8. The lack of coincident retrievals between the two sites on these dates is primarily attributed to persistent cloud cover in the region, which limited the number of valid sky radiance measurements. Although the inversions from Saint-Denis on October 19 satisfy the $AOD_{440} > 0.4$ and $\theta_S > 50°$ criteria (including two hybrid and one almucantar scans), they do not achieve Level 2 status due to insufficient valid radiances under the symmetry criteria, likely a consequence of the prevalent cloud cover (Level 2 inversions require more valid radiance angles than Level 1.5, for more details see Table 3 from Holben et al., 2006). For the retrievals from the R.V. Marion Dufresne, $R_{Vf}$ values ranging between 0.14 and 0.18 $\mu m$ and $R_{eff}$ values between 0.19 and 0.28 $\mu m$ are consistent across the inversions, align with the values observed on October 12, and agree with the retrievals from Saint-Denis on October 19. Notably, the previously discussed anti-correlation between fine-mode concentration ($C_{Vf}$) and real refractive index (RRI) is evident in the last three inversions: two from October 17 and, to a lesser extent, the final inversion from October 16. This phenomenon, as highlighted earlier, is a common retrieval artifact amplified under conditions such as limited angular radiance measurements caused by cloud cover. The first two inversions from October 16, despite having slightly lower solar zenith angles (SZA) than the Level 2 threshold of $50°$, appear to provide the most realistic retrievals for both fine-mode concentration and RRI. These retrievals show RRI values ranging between 1.43 and 1.50, consistent with the historical average RRI values for the nine Level 2 inversions from Saint-Denis, and align with expectations for a mixture of biomass burning and sea salt aerosols.

The single scattering albedo (SSA) values for the five R.V. Marion Dufresne inversions range between 0.88 and 0.95, with generally higher absorption observed at longer wavelengths. These values are consistent with the climatological SSA series for Saint-Denis, though they are slightly higher overall than the climatological averages mentioned earlier. As in the retrievals from October 12, the varying influence of sea salt aerosols in the mixture could explain the differences with the averaged Saint-Denis values. Additionally, Eck et al. (2013) reported a seasonal trend in SSA during the dry season at three South African sites, with values for pure biomass burning aerosols being highest at the end of the season. Apart from the variation on the sea salt contribution, this trend could also explain the differences with the averaged values, as the period analyzed here is October (while the nine inversions include also retrievals from August, September, and October). While some seasonality may exist, the SSA values retrieved from Saint-Denis on October 19 are notably higher and appear unrealistic for the observed mixture of biomass burning and sea salt aerosols during this week. This discrepancy is likely a result of cloud cover impacting the inversion quality, as retrievals that do not meet Level 2 criteria are expected to exhibit greater uncertainties.



## 5 Discussion

The development of the ship-adapted CIMEL 318-T photometer has been an ongoing process since 2017 within the framework of the Agora Lab, culminating in the establishment of the first permanent, operational and fully automated AERONET
ship-based site aboard the R.V. Marion Dufresne in July 2021, as part of the MAP-IO program. This milestone has enabled systematic aerosol monitoring in the Indian Ocean, yielding high-quality, AERONET-compatible shipborne spectral AOD measurements over a three-year period. These measurements keep the same protocols, calibration procedures, and data processing standards as AERONET ground-based photometers, ensuring consistency and compatibility.

The reliability of these AOD measurements has been demonstrated through intercomparisons conducted during the Amaryllis-
Amagas/Transama campaign, where two co-located shipborne photometers operated simultaneously aboard the R.V. Marion Dufresne, providing a unique opportunity to assess instrument consistency. These comparisons revealed strong correlations and low biases for AODs in Level 1.5 across all standard channels, comparable to those obtained at ground-based calibration sites during intercalibration periods with AERONET master instruments. Thus, the root-mean-square differences between the two photometers across all standard AOD channels were found to be lower than the expected AERONET error for differ-
ent wavelengths, estimated at 0.01 for standard channels and 0.02 for UV channels (Eck et al., 1999), further highlighting the system's accuracy. Similarly, comparisons with the fixed AERONET photometer at the Saint-Denis site on La Réunion, while accounting for environmental differences such as altitude and local urban influences, confirmed the reliability of shipborne measurements, showing high correlations. Both validation approaches indicate the robustness and quality of shipborne Level 1.5 data.

The two key technical implementations for obtaining accurate AOD measurements on shipborne platforms are real-time monitoring of the vessel's movements and continuous Sun (or Moon) tracking during measurements. As further described in subsection 2.1, the Trimble ABX-Two system provides precise attitude information, including pitch, roll, and heading, with an estimated error of less than $0.1 - 0.2°$ (considering the characteristics of our installation), allowing accurate determination of the Sun's position in dynamic conditions. Once the Sun enters the tracking system's field of view, the photometer switches
to a continuous tracking mode and maintains alignment throughout the whole direct Sun (or Moon) measurement. This is an adaptation specifically designed for this deployment, as ground-based CIMEL photometers perform tracking only once before each direct Sun (or Moon) triplet. This dual implementation ensures a high frequency of quality triplets that meet AERONET Level 1.5 criteria, as demonstrated in this study for large research vessels such as the R.V. Marion Dufresne, where ship movements are generally small and slow in calm atmospheric conditions.

To illustrate this, Figure 9 presents histograms of pitch (left panel) and roll (central panel) values measured over a 7-minute period during the almucantar measurement at 11:07 UTC on October 16. The dashed black line represents the mean pitch and roll values, which are annotated within the figure, while the black arrows indicate their respective standard deviations, also labeled. A normal distribution curve (black line) is fitted for reference. The values are displayed as a shaded histogram with black edges for clarity. These histograms represent typical conditions under which these measurements can be performed, with



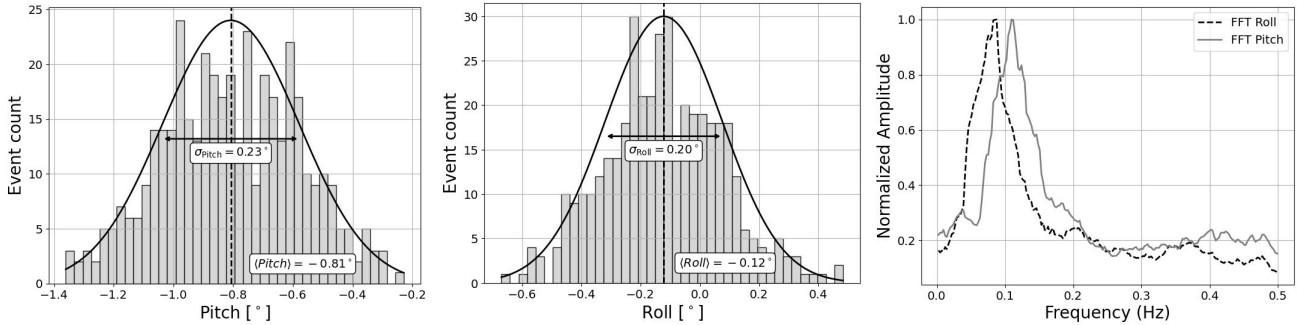

**Figure 9.** Histograms of pitch (left panel) and roll (right panel) values recorded during the 7-minute almucantar measurement at 11:07 UTC on October 16, 2023. The dashed black lines indicate the mean values, with corresponding standard deviations represented by black arrows. A normal distribution curve (black line) is fitted for reference. Mean values and standard deviations are also displayed in separate text boxes within each panel. The right panel presents the frequency spectra of these same pitch and roll movements, obtained via a Fast Fourier Transform (FFT).

mean pitch and roll values below $1°$ and standard deviations generally under $0.5$–$0.6°$ around these averages.[9]. Additionally, the right panel of Figure 9 presents the frequency spectra of these same pitch and roll movements, obtained via a Fast Fourier Transform (FFT), which decomposes the time series into its frequency components. A filtering step was applied to remove the zero-frequency component to focus on oscillatory behavior. The dominant frequencies are found around $0.1$ Hz (slightly lower for roll and slightly higher for pitch), corresponding to characteristic oscillation periods of approximately 10 seconds. This

means that vessel movements are not only small in magnitude but also slow in nature, further supporting the stability of the observational platform under calm conditions.

     A preliminary analysis of AOD data from similar new installations in 2024 on other large research vessels, such as the R.V. Gaia Blu (83 meters in length, as part of the ESA IDEAS-QA4EO program in collaboration with the Italian Istituto di Scienze Marine del Consiglio Nazionale delle Ricerche) and the R.V. Sarmiento de Balboa (72 meters in length, in collaboration

with Valladolid University and the Spanish Consejo Superior de Investigaciones Científicas), suggests that the system is well-adapted and effective for marine aerosol monitoring on large and stable vessels with movement characteristics similar to those observed aboard the R.V. Marion Dufresne, where AOD measurements at Level 1.5 have been routinely obtained. While data from these new installations have not yet been analyzed in detail, they hold significant potential for expanding the scope of marine aerosol monitoring using the system presented in Figure 1.

---

[9]The conditions depicted Figure 9 appear relatively calm, with the standard deviation being around $0.2°$ for both pitch and roll. However, a broader statistical analysis over a full year, using 7-minute averages of pitch and roll and the estimations of the corresponding standard deviations, indicates that this specific case falls within the lower 30th percentile of observed variability of the standard deviations. Over the annual dataset, the median (50th percentile) of the 7-minute standard deviation reaches approximately $0.43°$ for pitch and $0.58°$ for roll, which align with the general limits indicated here. These statistics account for all weather conditions, including both calm and stormy days. In clear-sky conditions, as typically required for direct Sun/Moon measurements, the percentile distribution is expected to shift toward lower values, as smaller standard deviations are generally associated with more stable atmospheric conditions.



However, despite its effectiveness on large and stable vessels, the system faces significant challenges when applied to smaller research vessels. Tests conducted on the NOAA R.V. Shearwater (a 20-meter high-speed catamaran) in September 2024 during the PACE-PAX campaign (more details in https://espo.nasa.gov/pace-pax) revealed that this vessel frequently experienced pitch and roll movements ranging between 5–10° and faster, even in clear conditions, causing the continuous tracking system to fail and preventing stable Level 1.5 AOD measurements. As a result, obtaining Level 1.5 AOD measurements under these

highly dynamic conditions proved challenging, highlighting the limitations of the standard combination of the Trimble system and continuous tracking for smaller vessels. To address this issue, a stabilizing mount, which actively counteracts the ship's movements, was tested while maintaining the continuous tracking system. This approach successfully enabled Level 1.5 AOD measurements, achieving a frequency comparable to that of larger vessels. However, while these initial tests have been promising, stabilizing mounts are significantly more expensive— approximately four to five times the cost of the Trimble system.

Additionally, their long-term durability in harsh maritime environments remains unproven, unlike the Trimble system, which has operated reliably for three years aboard the R.V. Marion Dufresne without issues. More testing is needed to assess whether stabilizing mounts can provide a sustainable and reliable solution for highly dynamic platforms over extended deployment periods. Another possible approach to improving stability on highly dynamic platforms is to develop a faster and adaptive tracking system within the Agora Lab, incorporating Proportional-Integral-Derivative (PID) control algorithms that dynamically adjust

tracking speed based on ship motion in pitch and roll. This adaptation would help maintain Sun tracking even under highly dynamic conditions, ensuring stable AOD measurements. Finally, it is important to mention that the Trimble ABX-Two system is currently experiencing stock availability issues, which may pose a significant challenge. Although it is a key component of the system architecture presented in Figure 1, it can be replaced with alternative market solutions, which are available and currently being explored to ensure a similar level of attitude precision while maintaining a cost-effective approach.

The capabilities of the shipborne photometer extend beyond automatic AOD measurements, as demonstrated by the first AERONET standard aerosol retrievals presented in section 4, conducted from the R.V. Marion Dufresne. These retrievals were made possible by the routine sky measurements performed onboard. While the pristine conditions of the southern Indian Ocean limited the number of quality-assured inversions (based on AERONET Level 2.0 retrievals criteria), those successfully retrieved during the study period from 12 to 19 October 2023 represent a significant breakthrough. These preliminary results primarily

serve to demonstrate the feasibility of conducting AERONET-type inversions from a moving platform. Inversions conducted in the port area (October 12) and at sea (October 16–17) confirm the capability of deriving detailed aerosol properties, including fine-mode modal radius and single scattering albedo (SSA), from shipborne platforms. These results align with those from fixed AERONET sites, despite the inherent challenges of measurements on a moving platform. The fine-mode modal radius remained consistent and the SSA values demonstrated high reliability, underscoring the potential of the system to deliver

accurate aerosol characterizations.

Although ship movements are recorded, with the Trimble ABX-Two system generating and storing positioning data every second, the current approach has been to invert radiance measurements using the standard almucantar angles without applying any corrections. The main reason for this choice is that the observed ship movements are generally small as shown in Figure 9. In this regard, a mean pitch and roll value below 1° falls within the leveling precision of standard ground-based AERONET





stations, as the bubble-level system of the CIMEL 318-T introduces a comparable uncertainty in such installations[10]. Additionally, the observed standard deviations are not much higher than the manufacturer's specified error for the Trimble ABX-Two system (0.1-0.2°), further questioning both the necessity and the potential accuracy of applying these angular corrections. Moreover, any post-processing correction would disrupt the symmetry criteria used in almucantar and hybrid scans, where left and right branch measurements are traditionally considered symmetric for cloud screening purposes. Given these factors, the

most appropriate use of the recorded ship attitude data appears to be as a quality control criterion, requiring mean pitch and roll values to remain below $\pm 1°$ and standard deviations below $0.5°$.

These standard deviation values of the ship's movements could be considered comparable to pointing errors in almucantar measurements. The impact of pointing errors on retrieval properties has been analyzed in previous studies. For instance, Dubovik et al. (2000) modeled a $1°$ mispointing in azimuth (twice the threshold suggested here) and found that the only signif-

icant effect on inversion products was observed in desert dust retrievals. It is worth noting that the authors did not average the left and right sides of the almucantar (personal communication), so the actual effect would likely be smaller. In a later study, Torres et al. (2014) tested azimuth and zenith errors of $0.4°$ (while also averaging almucantar measurements) and observed minimal influence on almucantar retrievals, even for desert dust. The largest effects were seen in the principal plane scenario, which, unlike almucantar scans, lacks the possibility of averaging and is now obsolete.

The analysis of the heading angle differs slightly from that of pitch and roll. This angle represents the ship's direction and is corrected each time the instrument performs Sun pointing and tracking. For shipborne photometers, this process occurs before each branch of the almucantar measurement seven wavelengths per branch, left and right), leading to a tracking update approximately every minute. The heading standard deviation observed within one-minute intervals are about half as large as those seen in pitch and roll, suggesting that their impact on almucantar symmetry is even less significant.

Further improvements to mitigate the impact of ship movement on sky radiance measurements could involve actively correcting the photometer's orientation in real-time using pitch and roll information from the attitude system. This approach appears to be a promising long-term and cost-effective solution, as it would eliminate the need for additional mechanical stabilization. However, its implementation would require significant development efforts from the instrument manufacturer, including the integration of dynamic motion compensation algorithms directly into the tracking system. Alternatively, a stabilizing platform,

like the one tested on the NOAA R.V. Shearwater, could be used to counteract ship movements and maintain stable pointing. This method is currently being tested and offers the advantage of simplifying post-processing by minimizing the need for attitude corrections. However, it also requires the integration of a costly stabilizing system, which may not be practical for all shipborne deployments.

The primary objective of this study was not to conduct an exhaustive analysis of aerosol inversions but rather to demonstrate

the feasibility of performing such measurements and retrieving aerosol optical properties from a moving platform in open ocean conditions. A more detailed validation could not be achieved with the R.V. Marion Dufresne dataset, as only one week with

---

[10]While the CIMEL 318-T is equipped with an internal bubble level, AERONET generally recommends using external leveling systems with higher precision, often achieving level adjustments of 4 mm per meter, corresponding to approximately $0.25°$. However, this recommendation is not always implemented in practice.



$AOD_{440} > 0.4$ was recorded over a three-year period, and even then, frequent cloud cover further limited the number of quality-assured retrievals, making a more comprehensive analysis unfeasible. Nonetheless, the successful retrievals obtained confirm the potential of shipborne aerosol inversions and establish a foundation for future, more detailed studies. In this regard, the dataset from the R.V. Gaia Blu, which has been operating in the Mediterranean Sea since February 2024, appears to be a strong candidate for a more extensive assessment. A preliminary evaluation of the data recorded by the shipborne photometer onboard R.V. Gaia Blu indicates a significantly higher occurrence of high-AOD aerosol events under clear-sky conditions, including episodes influenced by anthropogenic sources, desert dust, and even volcanic emissions from Etna. Moreover, the frequent proximity of the R.V. Gaia Blu to several Italian ground-based AERONET sites provides an excellent opportunity for a thorough validation of aerosol retrievals by enabling direct comparisons with the same retrievals performed using fixed photometer data. This will allow for an exhaustive assessment of retrieval performance, offering deeper insights into the accuracy and reliability of shipborne aerosol inversions, as well as the influence of ship motion on retrieval quality and potential limitations.

The installations in 2024, along with future deployments foreseen for 2025, will establish a foundation for a new network of ship-based automatic photometers. This network is expected to significantly enhance the study of aerosols in marine environments, particularly in regions that are otherwise challenging to monitor. In addition to supporting systematic aerosol characterization in purely maritime settings, this network will contribute to the validation of satellite observations. It will provide valuable ground-truth data not only for AOD product but also for more complex aerosol optical properties, such SSA. These data will be critical for validating the advanced retrievals from upcoming multi-angular polarimetric sensors onboard space missions, such as the ESA-EUMETSAT 3MI/MetOp-SG and NASA's PACE mission, which includes the SpexOne and HARP2 sensors (details and other mission sensors in Dubovik et al., 2019).

## 6   Conclusions

This study demonstrates the feasibility of establishing a permanent AERONET site aboard research vessels, adhering to the same automated measurement protocols and standards as conventional ground-based sites. Over a three-year period (July 2021–June 2024), the first ship-based AERONET site equipped with automatic photometer measurements was operated aboard the R.V. Marion Dufresne, primarily in the southwestern Indian Ocean, collecting a continuous AOD dataset as part of MAP-IO program. The mean AOD values—$0.093 \pm 0.075$, $0.063 \pm 0.041$, and $0.054 \pm 0.031$ at 440, 675, and 870 nm, respectively— are consistent with previous studies characterizing aerosols in uncontaminated marine environments. Similarly, the average Ångström Exponent (440/870) value of $0.76 \pm 0.40$ reflects the dominance of coarse-mode aerosols, with episodic contributions from fine-mode aerosols.

The reliability and precision of the ship-adapted CIMEL 318-T photometer were validated through detailed intercomparisons, particularly during the Amaryllis-Amagas/Transama campaign. During this campaign, two photometers (#1273 and #1243) were simultaneously operated aboard the R.V. Marion Dufresne between April and June 2023, enabling a direct comparison of their measurements. Strong correlations (R > 0.965) were observed across all analyzed AOD wavelengths (380–870 nm), with root-mean-squared errors (RMSE) ranging from 0.005 to 0.008, and biases between $-8.2 \times 10^{-5}$ and





$4.2 \times 10^{-3}$. These results confirm the high precision of the shipborne system and are comparable to those reported by Yin et al. (2019), who analyzed CIMEL measurements against MAN Microtops-II observations during the OCEANET campaigns aboard the R.V. Polarstern, finding RMSE values of 0.009–0.015 (their Figure 4). Notably, the strong correlations and minimal biases observed in AOD comparisons during this study align with those typically obtained at ground-based calibration sites during intercalibration periods with AERONET master instruments. This consistency underscores the reliability of shipborne

Level 1.5 data, reinforcing its comparability to well-established AERONET standards.

In addition to co-located comparisons, the shipborne photometer was validated against the AERONET ground-based site at Saint-Denis, located 93 meters above sea level and approximately 20 km from the port of La Réunion. A systematic negative bias was observed in the shipborne AOD data relative to Saint-Denis, ranging from $-7.0 \times 10^{-3}$ to $-1.3 \times 10^{-2}$, which can be attributed to altitude differences and variations in local aerosol sources. While Saint-Denis is considered a relatively clean

urban site, the R.V. Marion Dufresne is frequently docked in Le Port, an area influenced by industrial emissions, including those from a coal-fired power plant. Similarly, the Ångström Exponent exhibited a positive bias of 0.08, likely reflecting a greater contribution of fine-mode aerosols at Saint-Denis, where the reduced influence of marine aerosols allows urban aerosol contributions to become more dominant (Duflot et al., 2022). Despite these systematic differences, the strong correlations observed between the shipborne and ground-based measurements (R = 0.86–0.93) demonstrate that both instruments capture

similar temporal aerosol variability, further supporting the reliability of shipborne photometer.

A key achievement of this study is the successful application of the AERONET aerosol inversion algorithm to shipborne platforms, resulting in the first quality-assured retrievals of aerosol microphysical and optical properties from a moving research vessel. The data obtained during the biomass burning event of October 2023 exemplifies the system's capability to perform advanced aerosol characterization from remote marine observations. The spatial and temporal evolution of aerosol optical depth

(AOD) and Ångström Exponent during this event, as presented in Figure 6, highlights the distinct shift in aerosol characteristics, transitioning from pristine marine conditions to a fine-mode-dominated regime driven by biomass burning emissions. Between October 12 and October 19, $AOD_{440}$ values consistently exceeded 0.36, corresponding to the top $2.1\%$ percentile of the three-year dataset (approximately 580 observations). The highest $AOD_{440}$ value recorded during this week was 0.73, the maximum observed across the entire dataset. Weekly averages for this period were 0.45 for $AOD_{440}$ and 1.56 for the Ångström Exponent,

the latter significantly exceeding the 95th percentile value of 1.46, indicating a predominant influence of fine-mode aerosols transported from Madagascar, as confirmed by backward trajectory analyses.

Retrievals of aerosol microphysical and optical properties during this period from measurements aboard the R.V. Marion Dufresne were presented alongside those from the Saint-Denis site in Figures 7 (for the coincident day, October 12, at both sites) and 8 (for non-coincident days between the two sites). The retrieved fine-mode modal radii ($R_{Vf}$), summarized in Table 3,

remained remarkably consistent across inversions from both the shipborne platform and the Saint-Denis site, ranging from 0.13 to 0.18 $\mu$m. This consistency underscores the ability of the shipborne photometer to accurately characterize fine-mode particle size during biomass burning episodes. However, fine-mode volume concentrations ($C_{Vf}$) derived from the shipborne photometer were consistently higher than those retrieved at Saint-Denis. This discrepancy can be attributed to the well-documented



anti-correlation between fine-mode volume concentration and refractive index (Herrera et al., 2022), which may be further
amplified in inversions where limited angular radiance data, due to cloud cover, influences retrieval accuracy.

As noted by Sinyuk et al. (2020), even in cases with marked anti-correlation between fine-mode volume concentration and
refractive index, the AERONET retrieval algorithm reliably distinguishes between aerosol absorption and scattering proper-
ties. Indeed, the single scattering albedo (SSA) retrievals from the shipborne platform exhibit robust performance, with values
ranging from 0.88 to 0.95, indicating increased absorption at longer wavelengths. These SSA values are consistent with ex-
pectations for a mixture of biomass burning and sea salt aerosols (Dubovik et al., 2002b). Moreover, the SSA values obtained
during this week closely match with historical data from the Saint-Denis site under high aerosol load conditions, meeting the
$AOD_{440} > 0.4$ threshold required for AERONET Level 2 criteria.

In conclusion, the deployment of a CIMEL 318-T photometer aboard the R.V. Marion Dufresne has demonstrated its viability
for providing high-quality aerosol measurements comparable to those of AERONET ground-based stations. These findings
validate the system's potential to close observational gaps in marine aerosol research, offering a novel approach for satellite
validation and advancing our understanding of aerosol-climate interactions in remote oceanic environments.

*Data availability.* The AOD data from the shipborne photometer are publicly accessible at https://mobile.photons.univ-lille.fr/. The AOD
data from the Saint-Denis AERONET station, as well as the retrievals from the AERONET standard aerosol algorithm for this site, are
available at https://aeronet.gsfc.nasa.gov/. The aerosol retrieval data obtained from the shipborne photometer inversions are not currently
available online.

*Competing interests.* The contact author has declared that none of the authors has any competing interests.

*Acknowledgements.* This work was supported by the ESA-funded project QA4EO (Quality Assurance Framework for Earth Observation), the
EUMETSAT-funded project FRM4AER (Fiducial Reference Measurements for Copernicus Aerosol Product Cal/Val Activities), the CNES
(through the projects EECLAT, AOS and EXTRA-SAT), the European Union through H2020-INFRAIA-2014-2015 (ACTRIS-2, grant no.
654109) and Horizon Europe Projects: REALISTIC (grant no. 101086690) and the Marie Sklodowska-Curie Staff Exchange Actions with
the project GRASP-SYNERGY (grant no. 101131631). MAP-IO was funded by the European Union through the ERDF programme, the
University of La Réunion, the SGAR-Réunion, the Région Réunion, the CNRS, IFREMER, and the Flotte Océanographique Française. This
research has also been supported by the Ministerio de Ciencia e Innovación (grant no. PID2021-127588OB-I00) and by Junta de Castilla
y León with FEDER funds (CLU-2023-1-05). The technical developments of the shipborne photometer are part of the joint laboratory
AGORA-LAB (a collaboration between LOA and CIMEL Electronique).

The authors highly acknowledge the support of TAAF, IFREMER, LDAS, and GENAVIR for their assistance in the installation and
maintenance of scientific instruments aboard the R.V. Marion Dufresne. Special thanks are also extended to the technical teams of LACy and
OSU-R for their efforts in data acquisition and instrument maintenance.





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
