# Peer review of "Adaptation of the CIMEL-318T to Shipborne Use: Three Years of Automated AERONET-Compatible Aerosol Measurements Onboard the Research Vessel Marion Dufresne"

_EGUsphere, 2025_

## Author Comment (AC2)

**Answer to referee 1**

*General comments*

*The work by Torres et al. aims to demonstrate the feasibility of fully automated sun-photometer measurements aboard ships that meet AERONET standards, ensuring that shipborne data is consistent with existing land-based AERONET observations. This advancement supports the development of a shipborne AERONET-compatible network, addressing current observational gaps in aerosol measurements over remote maritime regions, enabling reliable assessments of aerosol optical depth measurements and other aerosol-related properties. The study details the adaptation of the CIMEL CE318-T Sun photometer for shipborne autonomous operation, and analyses data collected over a three-year period in the southwestern Indian Ocean aboard R.V. Marion Dufresne. The aerosol optical depth is validated through intercomparisons with co-located instruments and the nearby Saint-Denis AERONET site, and the first shipborne quality-assured AERONET aerosol retrievals are presented.*

We thank Referee 1 for their thorough and constructive review of our manuscript, as well as for the positive evaluation of its scientific significance and suitability for publication in Atmospheric Measurement Techniques. We appreciate the opportunity to address the comments, which have helped us to improve the clarity and structure of the manuscript. Below we provide detailed responses to each point, with changes implemented in the revised version of the manuscript.

*The manuscript is quite dense, with a large amount of background information provided in the early sections. While this provides useful context, it would benefit from streamlining and improved organization to enhance clarity and readability.*

We agree with this observation and have streamlined the Introduction to enhance clarity. In particular, we have moved the historical narrative describing the development of the sea-adapted CE318-T photometer from the end of the Introduction to the beginning of Section 2.1, where it more logically belongs. This allows the Introduction to focus more directly on the scientific context and motivation of the study, while improving the overall readability.

*Additionally, I suggest including a systematic cost-benefit analysis (space and power requirements, maintenance demands, personnel needs, etc.) and explicitly address the operational feasibility of broader deployment, useful to assess scalability.*

We thank the reviewer for this constructive suggestion. While a formal cost-benefit analysis remains beyond the scope of the present study, we have included a detailed paragraph addressing the system's operational feasibility in Section 2.1.2, "Current version and implementation". This addition discusses key aspects relevant to scalability, including space and power requirements, autonomous operation, minimal maintenance and personnel needs, and practical considerations for shipboard installation. We believe this information is best situated within the technical description of the instrument and its deployment, where it naturally complements the system overview.

*Overall, I consider the study to be scientifically significant and well-aligned with the scope and objectives of Atmospheric Measurement Techniques (AMT), therefore I recommend publication after minor revisions.*

*Specific comments*

*The introduction is generally well-written, effectively establishing the scientific context and motivation for the study. However, it presents a lot of foundational information before transitioning to the study's focus. A more direct introduction to the specific objectives of the research would make the introduction more engaging and accessible to a broader audience. For example, the detailed discussion of system configurations and preliminary tests aboard various research vessels, while informative, could be more effectively integrated later in the manuscript. Consider summarizing this content in the introduction and relocating the technical details to either subsequent sections or a new dedicated section. Also, the final part of the introduction could benefit from a more explicit presentation of the research objectives.*

We thank the reviewer for this helpful observation. This comment has also been raised by the other referees and the editor. In response, we have revised the structure of the manuscript to better reflect this recommendation. The detailed narrative on the system's development and early tests has been removed from the Introduction and is now presented in a dedicated subsection (Section 2.1.1, "System adaptation and historical development"), where it naturally follows the technical context of the study. In the Introduction, this content has been reduced to a brief summary, which improves the focus and flow of the section.

Furthermore, the final part of the Introduction has been rewritten to more clearly and explicitly present the main objectives of the study, highlighting the scientific goals and the relevance of shipborne AERONET-compatible measurements. We believe these changes enhance the readability and accessibility of the manuscript while preserving the necessary technical context.

*In section 2.2, I suggest improving the structure by breaking it down into clearer subsections corresponding to key stages (e.g., calibration, cloud screening, and quality control) to enhance readability. Additionally, consider reducing redundancy, as some of the information presented in section 2, overlaps with content in the Introduction.*

We thank the reviewer for this helpful suggestion. In response, we have restructured Section 2.2 under the title "Data Processing and Availability", introducing five dedicated subsubsections that reflect the key stages of the data workflow:

- 2.2.1. Acquisition, calibration and treatment
- 2.2.2. Correct functioning and Level~1.0 assignment
- 2.2.3. Cloud screening and Level~1.5 assignment
- 2.2.4. Sky radiances and inversion
- 2.2.5. Availability

This revised organization improves the overall logic and readability of the section by aligning with the natural sequence of data acquisition, processing, and dissemination. Additionally, some paragraphs have been relocated or slightly condensed to avoid redundancy with the Introduction and to enhance narrative flow. We believe this structure makes the content more accessible to the reader while preserving all essential technical details.

*In Section 3.1, the AOD averages presented appear to correspond to the complete dataset of valid measurements. In Section 3.2, it is stated that from April to June 2023, the R.V. Marion Dufresne was operating along the Brazilian coast. Were these data included in the averages shown in Tables 1 and 2? If so, the discussion regarding comparisons of average AOD conditions over the Indian Ocean with other studies should be revised, or the averages in the tables recalculated to exclude these data. Additionally, were there any other periods during the campaign when the vessel operated in regions outside the Indian Ocean? Please clarify this.*

We thank the referee for raising this important point. Indeed, the averages presented in Tables 1 and 2 were computed from the full dataset of valid AOD measurements collected aboard the R.V. Marion Dufresne between July 2021 and June 2024. This includes a short period (April–June 2023) during the Amaryllis-Amagas and Transama campaigns, when the vessel was relocated from its usual operational area in the southwestern Indian Ocean to the Brazilian coast and back.

However, this segment represents less than 5 % of the total dataset, and no photometer data were collected near the Brazilian coast due to a lack of authorization from the local authorities. Therefore, the available data for that period correspond exclusively to the transit across the South Atlantic between La Réunion and Brazil and back, preserving the consistency of observations within a remote marine environment. Moreover, we recalculated the averaged values using only the measurements from the Transama transit campaign ($AOD_{440}$= 0.08 and Ångström exponent = 0.06), and found them to be very similar to the global averages. This confirms their representativeness and minimal influence on the overall results.

Nevertheless, we have added an explicit clarification in Section 3.1 to avoid any ambiguity in the interpretation of the spatial coverage of the dataset.

*Section 3.2 could benefit from further use of subsections. For instance, it could be subdivided into parts that separately present the instrument consistency analysis (intercomparison between sun photometers #1273 and #1243) and the comparison with the Saint-Denis AERONET site.*

We thank the reviewer for this constructive suggestion. Following the recommendation, Section 3.2 has been subdivided into two subsubsections to clearly separate the two types of validation presented. The first, entitled *"3.2.1 Validation during Transama campaign"*, focuses on the intercomparison between the two shipborne photometers (#1273 and #1243) during the Transama campaign. The second, entitled *"3.2.2 Validation against AERONET Saint-Denis observations"*, presents the comparison between shipborne AOD measurements and those obtained at the ground-based AERONET site in Saint-Denis. This restructuring improves the clarity and readability of the section.

*The systematic cost-benefit analysis mentioned above could be included in section 5. Additionally, detailing the unique challenges associated with shipborne sun-photometer measurements in maritime environments would enhance this section.*

As mentioned in our response above, a brief evaluation of operational feasibility and scalability has been included in Section 2.1.2, "Current version and implementation", where the system is technically described and its real-world deployment is addressed. We consider this a more suitable location for such content, as it directly relates to the physical characteristics and practical use of the instrument. In contrast, we prefer to reserve the Discussion section for scientific and methodological aspects with implications for data interpretation or broader measurement strategies.

*Technical corrections*

*- Lines 164 to 168: there's some problem with the text here, please check and correct.*

Thank you for pointing this out. We believe this issue resulted from a text rendering problem during the PDF conversion process. In the version we downloaded from AMT, this section displays correctly. We expect the issue to be resolved in the revised version.

*- The Ångström exponent is mentioned several times, but the wavelength range used for its calculation is not clearly stated. I infer that it is 440–870 nm, as in the MAN dataset, but this should be explicitly specified in the manuscript. Consider including this information also in Tables 1 and 2, in the same way the AOD wavelengths are indicated.*

We thank the reviewer for pointing this out. The wavelength range used for the Ångström Exponent calculation is now explicitly stated in Section 2.2.1, where the data processing procedures are described. Specifically, we indicate that the 440–870 nm range is used throughout this study, as infered by the referee since it is the most commonly adopted range in the AERONET literature and widely used in related studies, including the MAN dataset. Since this definition is consistently applied across the manuscript, we do not include the wavelength range as a subscript (e.g. $\alpha_{440-870}$) in each occurrence, to avoid redundancy.

---

## Author Comment (AC3)

**Answer to referee 3**

*The study performed by B. Torres et al. presents three years of solar and lunar AOD measurements aboard a research vessel in the area of the Indian Ocean, using a Cimel CE318T automatic sunphotometer, and following the standard procedure kept by the Aerosol Robotic Network (AERONET). The analysis also includes for the first time the measurements of sky radiance, performed in the almucantar and also using hybrid escenarios. The results show good performance, comparable to standard measurements taken at ground sites from AERONET. It is therefore considered kind of foundational paper for the future establishment of a network of instruments deployed at vessels. This is an important obljective given the huge gap of data found in vast oceans. The text also points at future further developments in order to improve current limitations. The results are of scientific interest, well within the scope of the journal. The English usage is very good to my understanding, and it has been written and composed with care. However, I would recommend some major changes (on the structure mainly) before its acceptance.*

We thank Referee 3 for their positive assessment of our work and for the constructive comments, which have helped us improve the structure and clarity of the manuscript. Below we address each point in turn:

*General comments*

*The introduction is interesting and informative. However, I think many paragraphs should be moved to section 2. In fact, some of the information is redundant in section 2. Please, keep the introduction shorter, and integrate the removed paragraphs in section 2. I do not recomment elimination of information, but integration in the next section.*

We agree with the reviewer that some paragraphs initially placed in the Introduction overlapped with content in Section 2. Following all reviewer's suggestion, we have revised and streamlined the Introduction by transferring history and technical details to Section 2, where they are more appropriately discussed. At the same time, we have slightly shortened some parts to avoid repetition.

*I also think that section 5 (discussion) should be integrated in section 4 (results). In fact, the initial paragraphs in section 5 are redundant again. By moving section 5 to 4 you can eliminate them.*

We acknowledge the reviewer's comment regarding redundancy. The opening paragraphs of Section 5 have been substantially revised to remove repetition with earlier sections. While we have not fully merged Section 5 into Section 4, we have significantly condensed the discussion. We chose to retain a dedicated Discussion section to allow for a more interpretative and contextual analysis of the results without interrupting the flow of the main findings.

*It would be ilustrative to include an image of the new system and the platform in section 2.*

A detailed schematic of the system and its modular components is already included in the current version (see Figure 2). We believe this provides sufficient visual context regarding the system design

and installation. Additional photographs of the platform and its integration aboard the R.V. Marion Dufresne are available in the appendix Tulet et al. 2024, which is cited in the manuscript.

*Similar ship version developments of Prede POM instruments were tested in japanese R.V. Shirase. It would be interesting to cite as an example in the introduction. See for example Kobayashi et al. (DOI: 10.1117/12.2195691).*

We thank the reviewer for this useful reference. We have now included a citation to Kobayashi et al. (DOI: 10.1117/12.2195691) in the Introduction as an example of prior ship-based aerosol photometry with Prede POM instruments. This addition enriches the context and acknowledges other important efforts in this direction.

*Specific comments and corrections:*

*- Line 10: Angström exponent is writte differently in the text. Please correct them accordingly (for example, it appears incorrectly at line 10, 100, table 1 and 2)*

We thank the reviewer for pointing this out. The spelling of "Ångström exponent" has now been corrected and standardized throughout the manuscript. Starting from Section 2.2.1, we have adopted the symbol $\alpha$ to refer to the Ångström exponent as defined in that section, and this notation is consistently used throughout the rest of the text.

*- Line 54: I would say that "preindustrial" conditions is not the best term to use, as the earth is already affected by anthropogenic emissions, even in remote areas. Maybe using remote oceanic conditions, or natural background conditions would suit better?*

We thank the reviewer for this suggestion. While we understand the concern regarding the pervasiveness of anthropogenic emissions, we have retained the term "preindustrial" following the usage in Hamilton et al. (2014), which provides a quantitative definition of "preindustrial-like" regions based on aerosol properties and model simulations. As described in their study, a significant portion of the Southern Hemisphere oceans—particularly the tropical and mid-latitude areas—still exhibit aerosol conditions that closely resemble the preindustrial atmosphere, both in magnitude and behavior.

In this context, our use of "preindustrial conditions" refers to these rare, low-aerosol regions identified as baselines for quantifying anthropogenic forcing, and we believe it remains appropriate given the location of our measurements in the Southern Indian Ocean. Nonetheless, if the editor prefers a more neutral term such as "natural background conditions," we would be happy to adjust accordingly.

*- Line 101: Does AERONET use least-sqyares method over 440-870 nm wavelength range? Can you confirm? I thought the Angstrom exponent was calculated by using rations of channels 440 and 870 nm.*

We thank the reviewer for raising this point. In AERONET, the Ångström exponent is indeed calculated using a least-squares linear regression in log–log space of aerosol optical depth (AOD) versus

wavelength. This calculation is typically performed over specific spectral ranges, the most commonly used being 440–870 nm. The regression uses all available AOD measurements within the selected range, depending on data availability and quality at each processing level (e.g., Level 1.0, 1.5, or 2.0).

We have clarified this in Section 2.1.2 of the manuscript, which describes the data processing procedures. In addition, we have updated the description of the MAN dataset in the Introduction to explicitly state that the Ångström exponent there is also derived using a least-squares linear regression in log–log space.

*- Line 155: attempts*

The paragraph has been edited, corrected and moved to the next section.

*- Lines 164-167: there a series of typos and words sticked together that look caused by editor software problems: andber, imprimproved, greatlypared, shoshowing, squaredferences, all alls...*

We thank the reviewer for pointing out these issues. These errors appear to be artifacts introduced during PDF rendering or typesetting, as they are not present in the source manuscript.

*- Line 182-183: revise the sentence please.*

The paragraph has been edited, corrected and moved to the next section.

*- Line 194: Why not using a different name for the version of Cimel CE318-T?*

We thank the reviewer for the suggestion. At this stage, the shipborne photometer remains a prototype based on the standard CE318-T core, with adaptations specifically designed for autonomous marine operation. Since the system is still under development and not yet industrialized as a separate commercial product, we have opted to retain the original designation. However, we agree that a distinct name may be appropriate in the future.

*- Line 398: characteristic for the whole indian ocean or only SW?*

We thank the reviewer for the observation. The sentence has been clarified to specify that these conditions are characteristic of the southwestern Indian Ocean, which corresponds to the primary operational region of the R.V. Marion Dufresne. The manuscript has been updated accordingly.

*- Figure 3: why not merging together the two plots? Is there a problem in readibility?*

We thank the reviewer for the suggestion. We initially considered merging the two plots in Figure 3, but given the three-year timeline and the density of daily data points, combining both plots significantly reduced readability. Splitting the figure into two panels allowed for a clearer visual representation of the time series.

*- Line 484: Please add a reference for last sentence.*

We thank the reviewer for this observation. As also noted by another reviewer, a reference has now been added to support this statement. Specifically, we cite Holben et al. (1998), which reports that the total uncertainty in AOD from a newly calibrated field instrument is typically below 0.01 for wavelengths above 440 nm and below 0.02 for shorter wavelengths under cloud-free conditions. These uncertainty levels are based in part on root-mean-square differences observed during intercalibrations with AERONET reference instruments and are consistent with the AOD biases observed in our own intercomparison. Please note also that several authors of this study are directly involved in the calibration of a large number of AERONET instruments worldwide, and in this context, we confirm that these thresholds with respect to the master instrument are routinely applied as acceptance criteria for new valid calibrations.

*- Line 494-499: this paragraph is redundant.*

We thank the reviewer for the observation. The content originally flagged as potentially redundant has now been restructured and incorporated into Section 3.2.2, which is dedicated to the comparison with the AERONET Saint-Denis site. Within this new subsection, the paragraph serves to introduce and contextualize the comparison in a more concise and relevant manner. We believe this improves the organization of the manuscript.

*- Line 503: state kilometers as km, as done in other appearances.*

Thanks, corrected.

*- Line 529: It is a pity not to include a period in which the instrument was installed at ground in the port to estimate the differences with Saint-Denis, so effect of the vessel could be removed.*

We agree with the reviewer that having a reference period with the instrument installed at ground level near the port—prior to embarkation—would have allowed a more direct assessment of the vessel's influence on AOD measurements. Unfortunately, such a comparison was not feasible during this deployment, as no measurements were taken under these conditions. However, this is a valuable point that will be addressed in future studies. For instance, the R.V. Gaia Blu regularly operates from the Port of Naples, and its dock is located only 2 km from the AERONET site Napoli_CeSMA. This setup offers an excellent opportunity for land–sea cross-validation in upcoming studies.

*- Figure 6: The black cross symbol is difficult to find...*

We thank the reviewer for this suggestion. The black cross was chosen to ensure sufficient contrast against the red/orange background of the NASA Worldview AOD data. However, we understand that its visibility may vary depending on the viewing conditions or resolution. We will review this point during the final figure editing process and make adjustments if necessary, in coordination with the graphical editor, to improve clarity.

*- Figure 7: The two M.D.Alm colors used for left figure are somewhat difficult to distinguish, mainly in printed material.*

We thank the referee for this observation. Following the graphical editor's recommendation regarding Figure 8 and its accessibility for color vision deficiencies, we revised the color palette used in both Figures 7 and 8. The updated palette has been carefully selected to improve visual contrast and accessibility, ensuring distinguishability across all curves—even in printed versions or for readers with color vision deficiencies. We believe the figures are now significantly clearer and more informative.

*- Line 666-668: please review the sentence.*

We thank the reviewer for pointing this out. The sentence has been revised for clarity and correctness.

*- Line 696-697: Repeated sentence*

Thanks, the repetition has been erased.

*- Line 716: SSA already defined in 625*

Thanks, corrected.

*- Line 760: extra dot*

Thanks, corrected.

*- Footnote 9: It would be interesting to add the corresponding histograms for comparison.*

We thank the reviewer for the suggestion. While we agree that including the histograms could provide additional context, we believe the current figure and discussion already offer a sufficient level of detail for the intended comparison. Given the length of the discussion and the focus of the manuscript, we prefer to keep the analysis concise at this stage. However, we acknowledge that this could be an interesting addition in future studies or supplementary material.

*- Line 835-836: repeated*

We thank the reviewer for the remark. Upon careful review, we believe the current paragraph does not contain any actual repetition. Rather, it presents two alternative approaches (real-time orientation correction and mechanical stabilization) using a parallel structure to clearly contrast their respective advantages and limitations. We hope this clarifies the intent and are happy to revise further if the editor finds it necessary.

---

## Author Comment (AC4)

**Answer to referee 2**

*General comments*

*This work presents the results of what can be considered a great step forward in the use of photometry for the study and monitoring of atmospheric aerosol, extending to the marine domain what until now has been a technique possible only in fixed locations.*

*The work is very well written and clear, also from the point of view of English.*

We sincerely thank Referee 2 for the constructive and encouraging feedback, and we grateful for the recognition of the value of this work as a significant step toward extending high-quality photometric aerosol monitoring to the marine environment. We address below each of the comments in detail.

*My only concerns are the following:*

*- the introduction seems a bit too long containing perhaps too many details on the description of aerosols, their role in the climate, on measurement techniques. Furthermore, in the second part the history of the development of this system suitable for operating on ships is reconstructed. This information is then repeated in section 2 (where it is certainly more suitable)*

We thank the reviewer for this observation. While we have shortened and reorganized the second part of the Introduction—moving the historical account of the shipborne system to Section 2, where it is more appropriate—we have opted to retain the first part, which reviews the importance and characteristics of marine aerosols. This section was carefully written in collaboration with Dr. Alexander Smirnov, principal developer and coordinator of the Maritime Aerosol Network (MAN), with the aim of recovering key past contributions regarding marine aerosol research. We believe this overview provides valuable context not only for interpreting the present study, but also as a resource for new generations of researchers entering this field.

That said, we have reviewed the structure and wording of this part to ensure conciseness and focus, avoiding unnecessary digressions and overlap with Section 2.

*- section 2.2 reports in too much detail what is the standard analysis procedure of AERONET. I think it is enough to refer to the article that is in fact cited several times (Giles et al. 2019)*

We have carefully revised Section 2.2 and trimmed down parts of the explanation regarding standard AERONET processing steps. Nevertheless, we retain some key points to clarify the context and applicability of certain procedures in a shipborne context. These include explicit details about calibration, cloud screening, and Level 1.5 data assignment, as these are critical for validating and understanding the reliability of marine-based data and the differences from ground-based scenarios. All relevant sections now emphasize references to the main AERONET documentation, particularly Giles et al. 2019 for procedural details.

*- the results are illustrated in detail in section 3. It is therefore not clear to me the usefulness of repeating them in section 5*

We thank the reviewer for this remark, which has led us to revise Section 5. The preamble of the Discussion has now been substantially shortened to avoid repeating material from Section 3. Instead, we have focused on a more concise synthesis and interpretation of the findings in the context of previous literature and future applications.

*Specific comments:*

*L27 Which is the meaning of specifying the percentage of international waters?*

We agree this detail may be overly specific at this early stage of the paper. This sentence has been removed in the revised version to improve the flow and focus of the Introduction.

*L209 Not only the Sun, but also the Moon, so I would say "locked onto the target"*

All this section 2.1 has deeply evolved but we have kept on mind to add the Moon for the tracking issue.

*L222-223 What about sea spray?*

This part now is in 2.1.2 and we have added the airshield information to avoid sea spray as presented in figure 1.

*L230-231 It's not clear to me what this means "SUN, MOON, (Sun and Moon direct measurements)". Maybe the second comma is not necessary?*

We agree with the reviewer that the original phrasing was unclear. The terms "SUN" and "MOON" referred to internal CIMEL codes used to designate direct Sun and direct Moon observation scenarios. As these are not meaningful to general readers, we have removed these internal references from the manuscript to avoid confusion.

*L234 "identical to those applied at regular fixed ground-based sites". This concept has been repeated many times.*

We acknowledge the redundancy pointed out by the reviewer. The affected sentence has been revised and shortened to avoid repeating that the AERONET protocols are identical unless such emphasis is necessary—for instance, when highlighting specific adaptations for shipborne or moving platforms. Additionally, in several parts of the text we now use the term "AERONET-compatible" to streamline the discussion and avoid unnecessary repetition.

*L231-234 120+220 days doesn't make a full year*

We thank the referee for pointing out the discrepancy in the total number of operational days per year. In addition to the 220 days operated by Ifremer and the 120 days under TAAF, the remaining ~25 days per year correspond to periods when the R.V. Marion Dufresne is docked for port operations, vessel maintenance, and technical upgrades. These intervals are also used for maintenance and recalibration of the scientific instruments installed on board. This clarification has now been incorporated into the revised manuscript (Section 2.3).

*L264 Why 5% percentile is not reported as you did with 95%?*

Thanks for the remark, the 5% has been added.

*L401-407 Could you give more details or put a reference on the evaluation of indetermination of Angstrom Exponent at very low AOD?*

We thank the reviewer for this observation. As explained in the manuscript, the increased variability of $\alpha$ under very low AOD conditions is primarily due to the relative nature of the uncertainty: while AOD errors are generally considered absolute (e.g., 0.01 for standard channels), their impact on the $\alpha$ calculation becomes more significant as AOD decreases. This can be easily shown since $\alpha$ is derived from a least-squares fit in log–log space, where the error in log(AOD) mathematically increases for smaller AOD values.

More specifically, since the $\alpha$ is calculated from a linear regression of log(AOD) versus log($\lambda$), the uncertainty in $\alpha$ is influenced by the propagation of the relative error in AOD. As $\Delta[\log(\text{AOD})] = \Delta(\text{AOD}) / \text{AOD}$, the error contribution becomes larger as AOD decreases. This effect is particularly relevant for marine environments, where very low AOD values are common, especially in the near-infrared range.

We believe the current explanation in the manuscript provides a clear and sufficient account of this limitation, particularly for readers familiar with AERONET methodologies. However, should the editor consider a more formal treatment necessary, we would be happy to include additional clarifications or references in the revised version.

*L425-432 Is not clear to me which is the meaning of give all this numerical details (e.g. different percentiles) on the AOD and AE values during the BB event*

We thank the reviewer for this observation. The numerical details, including percentiles and maximum values, are provided to objectively characterize the intensity and uniqueness of the biomass burning event within the context of the three-year dataset. While the AOD values observed during this episode are moderate when compared to AERONET sites frequently affected by biomass burning, they are the highest in our dataset and clearly distinguishable from the background pristine marine conditions.

Importantly, this episode is not only statistically prominent but also temporally isolated: all elevated values are confined to that specific week, and there are no other days or isolated measurements with similarly high AOD or low Ångström exponent values throughout the entire dataset. This reinforces the exceptional nature of the event and justifies its selection for detailed analysis in Section 4, where inversion products are presented. The statistical description therefore supports both the identification and the scientific relevance of this unique case.

*L482-484 Could you provide a reference for this?*

We thank the reviewer for this observation. A reference has now been added to support this statement. Specifically, we cite Holben et al. (1998), which reports that the total uncertainty in AOD from a newly calibrated field instrument is typically below 0.01 for wavelengths above 440 nm and below 0.02 for shorter wavelengths under cloud-free conditions. These uncertainty levels are based in part on root-mean-square differences observed during intercalibrations with AERONET reference instruments and are consistent with the AOD biases observed in our own current intercomparison/intercalibration. Please note also that several authors of this study are directly involved in the calibration of a large number of AERONET instruments worldwide, and in this context, we confirm that these thresholds with respect to the master instrument are routinely applied as acceptance criteria for valid calibrations.

*L517 How is calculated this bias? I don't find it*

We thank the reviewer for the comment. As stated in the manuscript, the bias is calculated using the AERONET Saint-Denis site as a reference. It is computed as the mean of the signed point-by-point differences in AOD between both instruments. We preferred not to overload the text with additional technical details, but should the editor consider it necessary, we would be happy to add a clarification.

*L538 "aerosol retrieval" sounds too generic in my opinion. Maybe you can "optical-physical"?*

We thank the reviewer for this suggestion. We agree that "aerosol retrieval" could be interpreted as too generic. To improve clarity, the title of Section 4 has been updated to: "First quality-assured shipborne AERONET retrievals of aerosol optical and microphysical properties."

*L596 "exceeding the 95% percentile value of 1.46" of the total period?*

We thank the reviewer for this observation. Yes, the 95th percentile value of 1.46 refers to the Ångström exponent distribution over the entire three-year dataset from the R.V. Marion Dufresne. This clarification has now been included in the revised manuscript to avoid ambiguity.

*L595-598 In these sentences there are repetitions of AOD and SZA values.*

We thank the reviewer for this observation. However, upon careful review, we believe there is no repetition in the referenced sentences. The text describes two distinct SZA–AOD combinations observed during the biomass burning event. Each pair is presented to illustrate the range of conditions contributing to elevated Ångström exponent values. We hope this clarifies the intended structure.

*Typos:*

*L67 analyzes*

We thank the reviewer. "Analyzes" has been corrected to "analysis" in line 67.

*L94 platformS*

Added, thanks.

*L164 andber, imprimproved greatlypared*

Thank you for pointing this out. We believe this issue resulted from a text rendering problem during the PDF conversion process. We expect the issue to be resolved in the revised version.

*L167 for all alls*

This part has been completely edited and this sentence has been corrected.

*L170-171 "This solution was shown inefficient for our proposed automated independent final solution". Repetition*

This part has been completely edited and we have avoided the repetition.

*L186 I would suggest "identified" instead of "discovered"*

Ok, thanks for the suggestion, we have changed it.

*L190 I would write "the first instrument fully compatible with AERONET" or "the first fully AERONET compatible instrument"*

*L192 "marking a significant milestone in achieving 100% AERONET compatibility" is a repetition of what stated at line 190*

We thank the reviewer for these suggestions. To address the redundancy and improve clarity, we have revised the text accordingly. Specifically, we consistently use the term *AERONET-compatible* throughout the manuscript to simplify the expression and avoid repetition. The phrase in line 192 has also been shortened to eliminate redundancy with line 190.

---

## Author Comment (AC5)

*Dear corresponding author, dear authors,*

*As editor of this manuscript, I agree with all three reviewers, that the manuscript is valuable to be published in this journal.*

*It is of good quality, as well scientifically, and well written. The topic is of course inovative and of high valuable interest.*

*Nevertheless I also agree with the reviewers that some corrections have to be done. I summarize all the main recommandations that I found in reviewer comment 1 (RC1), reviewer comment 2 (RC2), reviewer comment 3 (RC3), and make following recommandations in a synthesis:*

Dear Editor,

We would like to sincerely thank you for your positive evaluation of our manuscript and for your support throughout the review process. We are grateful to all three reviewers for their constructive and detailed comments, which have helped us improve the clarity, structure, and scientific quality of the paper.

Following your synthesis and recommendations, we have carefully revised the manuscript as follows:

*The introduction is too long (RC1, RC2, RC3)*

In response to RC1, RC2, and RC3, the Introduction has been significantly shortened. The historical instrumental development has been moved to Section 2, where it is more appropriately discussed.

*You should Shorten Section 2 with less details about aeronet procedure (RC2).*

In line with RC2's suggestion, we have reduced the level of detail concerning standard AERONET procedures, keeping only the essential points needed to understand the processing of our dataset. We now refer readers to Giles et al. (2019) for complete methodological details. In addition, we have reorganized the content into distinct subsections to improve readability and guide the reader more easily through the various processing steps.

*Please restructure to shorten and avoid redundancy:*

*- Restructuration introduction and section 2 (RC3, RC2) -> Shorten Intro and part of them to section 2*

*- Restructuration Section 3 (data analysis) & 4 (results) and Section 5 (conclusions) (RC2, RC3)*

We have restructured Section 3 to include new subsections where appropriate (e.g., the intercomparison between instruments and the validation against the Saint-Denis site, as suggested by RC1). We have

kept Section 5 (Discussion) but significantly reduced its opening paragraphs to eliminate redundancy with sections 3 and 4, as suggested by RC2 and RC3.

*I suggest as the reviewers did these significant changes/additions:*

*-> Also mention the Prede developments in the introduction (RC3)*

Following RC3's suggestion, we now mention the development of ship-adapted Prede POM instruments, including a reference to Kobayashi et al. (2014), in the Introduction.

*-> Ångström Exponent should be properly defined (RC1, RC2, RC3) -> On which wavelengths/channels is it computed: only two wavelengths (limit wavelengths of the range of computation)? All the channels of the wvelength range?*

As requested by all three reviewers, we now explicitly define the Ångström Exponent as computed by least-squares linear regression in log–log space over the 440–870 nm range. This is stated in Section 2.2.1 and mentioned also in the Introduction and in the caption of Tables 1 and 2. Throughout the manuscript, we refer to it as $\alpha$ once defined, for clarity and to avoid repetition.

*I have also some own recommandations:*

*-> For me it is allways very important to add a table of accronyms and maybe of parameters.*

We have included a new table listing all acronyms and key parameters as you recommended.

*-> At the end: Please replace "TEXT" with information by:*

*- Author contributions: "TEXT"*

*- Disclaimer: "TEXT"*

These sections have now been completed at the end of the manuscript.

*-> Consider all specific comments, corrections, typos and questions of RC1, RC2, RC3... And of course also an foremost the general comments of them that I have not mentionned above.*

Figures and color accessibility: Following the comment by RC1 and the journal's graphics editor, we revised the color palettes of Figures 7 and 8 to ensure accessibility for color-blind readers, using Coblis to verify the improvements.

All specific corrections and clarifications addressed: We have addressed all specific line-by-line comments, typos, and requests for clarification from RC1, RC2, and RC3.

*Have a lot of success in the (short) corrections of the manuscript, we are looking forward to validating the next version of it for final publication.*

We hope that the revised version of the manuscript now meets the standards for publication in Atmospheric Measurement Techniques. Please do not hesitate to contact us should any further clarification or adjustment be needed.

Thank you once again for your support and guidance throughout this process.

Warm regards,

Benjamin Torres

on behalf of all co-authors